# RAVENEA: A BENCHMARK FOR MULTIMODAL RETRIEVAL-AUGMENTED VISUAL CULTURE UNDERSTANDING

**Jiaang Li**[1*]    **Yifei Yuan**[1,2*]    **Wenyan Li**[1]    **Mohammad Aliannejadi**[3]
**Daniel Hershcovich**[1]    **Anders Søgaard**[1]    **Ivan Vulić**[4]
**Wenxuan Zhang**[6]    **Paul Pu Liang**[5]    **Yang Deng**[7†]    **Serge Belongie**[1†]

[1]University of Copenhagen    [2]ETH Zürich    [3]University of Amsterdam
[4]University of Cambridge    [5]Massachusetts Institute of Technology
[6]Singapore University of Technology and Design    [7]Singapore Management University

## ABSTRACT

As vision-language models (VLMs) become increasingly integrated into daily life, the need for accurate visual culture understanding is becoming critical. Yet, these models frequently fall short in interpreting cultural nuances effectively. Prior work has demonstrated the effectiveness of retrieval-augmented generation (RAG) in enhancing cultural understanding in text-only settings, while its application in multimodal scenarios remains underexplored. To bridge this gap, we introduce RAVENEA (**R**etrieval-**A**ugmented **V**isual cultur**E** u**N**d**E**rst**A**nding), a new benchmark designed to advance visual culture understanding through retrieval, focusing on two tasks: culture-centric visual question answering (cVQA) and culture-informed image captioning (cIC). RAVENEA extends existing datasets by integrating 11,396 unique Wikipedia documents curated and ranked by human annotators. Through the extensive evaluation on seven multimodal retrievers and seventeen VLMs, RAVENEA reveals some undiscovered findings: (i) In general, cultural grounding annotations can enhance multimodal retrieval and corresponding downstream tasks. (ii) VLMs, when augmented with culture-aware retrieval, generally outperform their non-augmented counterparts (by averaging $+6\%$ on cVQA and $+11\%$ on cIC). (iii) Performance of culture-aware retrieval augmented varies widely across countries. These findings highlight the limitations of current multimodal retrievers and VLMs, underscoring the need to enhance visual culture understanding within RAG systems. We believe RAVENEA offers a valuable resource for advancing research on retrieval-augmented visual culture understanding.

**Project page:** jiaangli.github.io/ravenea

## 1 INTRODUCTION

Vision-language models (VLMs) are increasingly deployed in real-world applications, from education to assistive technologies (Baker et al., 2021; Di Nuovo et al., 2024; De Marsico et al., 2024; Karamolegkou et al., 2025), where understanding not only visual content but also the surrounding cultural context is crucial. Despite achieving impressive performance on general tasks (Lin et al., 2014; Yue et al., 2024; Bai et al., 2025a), VLMs often struggle to capture cultural nuances, such as traditions, symbols, and region-specific practices that require external, culturally grounded knowledge (Nayak et al., 2024; Khanuja et al., 2024; Liu et al., 2025; Winata et al., 2025). For example, when analyzing images of cultural events, VLMs may misinterpret region-specific attire as generic clothing, thereby overlooking its ceremonial significance (Nayak et al., 2024). What's more, they also risk reinforcing existing cultural biases, as biased collections or metadata frequently lead VLMs to perform much better in mainstream culture contexts (e.g. western culture) while marginalizing

---

*Equal contribution.

†Principal senior advisor.

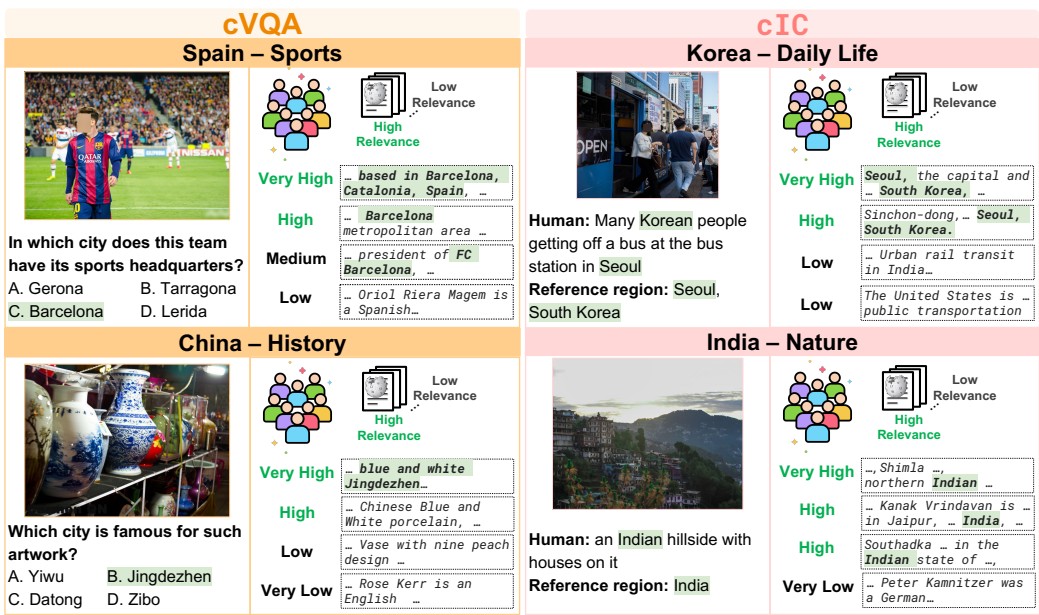

Figure 1: **Samples from RAVENEA.** RAVENEA evaluates the multimodal culture-awareness RAG with VLMs across eight countries spanning four continents: Asia, Africa, Europe, and Latin America. It comprises two tasks designed to assess visual culture understanding abilities and the effect of extra cultural knowledge across different countries. With 1,868 instances and 11,369 human-ranked documents, RAVENEA covers eleven real-world scenarios, including architecture, cuisine, art, etc.

minority traditions (Nayak et al., 2024). A promising approach to mitigate the limitations is the integration of external knowledge through retrieval-augmented generation (RAG) (Lewis et al., 2020), which has shown success in improving cultural awareness in language models (Seo et al., 2025; Lertvittayakumjorn et al., 2025). However, prior work on culture has largely remained in text-only settings, while existing multimodal culture-related datasets mostly probe VLMs' memorized cultural knowledge instead of testing their understanding of culture in real-world contexts. Consequently, **it remains unclear to what extent current mutlimodal retrievers support reliable cultural retrieval, and whether RAG can effectively advance multimodal cultural understanding for VLMs**. This shortfall underscores the need for a dedicated benchmark that probes various facets of visual culture comprehension for multimodal retrievers and VLMs.

To bridge the gap, we introduce RAVENEA (**R**etrieval-**A**ugmented **V**isual cultur**E** u**N**d**E**rst**A**nding), a *manually curated* dataset designed to evaluate multicultural understanding in VLMs with retrieval support. We construct RAVENEA based on two existing datasets: CVQA (Romero et al., 2025), which includes culturally relevant visual questions and corresponding answers, and CCUB (Liu et al., 2023), offering culturally contextualized captions to foster inclusivity in text-to-image generation.[1] For each instance drawn from the source datasets, we append a set of Wikipedia documents that have been **human-ranked** based on their cultural relevance to the associated image. This curation effort, designed to ensure broad cultural representation, contains data related to **eight** countries and spans **eleven** diverse categories (e.g. architecture, art, cuisine, etc), comprising **18,680 human-labeled** image-document pairs. RAVENEA thus provides a multimodal retrieval-augmented benchmark for evaluating multicultural sensitivity of various retrievers, and further allows for assessing how well VLMs utilize retrieved cultural context in their reasoning process. Specifically, as shown in Figure 1, we focus on two culturally grounded tasks: (i) **culture-centric visual question answering (cVQA)** and (ii) **culture-informed image captioning (cIC)**.

With RAVENEA, we first train and evaluate seven multimodal retrievers that use both visual and textual inputs to retrieve Wikipedia documents for a given image query based on their cultural relevance.

---

[1]We reuse the cultural captions from CCUB as ground-truth references for the inverse task, image-to-text generation, specifically for culture-aware image captioning.

We then evaluate 17 widely used VLMs, both with and without multimodal retrieval, spanning open-source and proprietary models as well as sizes from 2B to 78B parameters, to assess the effect of retrieval augmentation on cultural understanding in the cVQA and cIC tasks. Our benchmark provides a testbed for evaluating the cultural relevance capabilities of multimodal retrievers and the effectiveness of VLMs in consuming and using such retrieved cultural context.

Our contributions and key findings include:

- **RAVENEA benchmark**: We introduce RAVENEA, the first benchmark for evaluating VLMs in using external knowledge for visual culture understanding. It covers eight countries across eleven categories, linking images to **human-ranked** Wikipedia documents on two tasks: cVQA and cIC.
- **Cultural grounding annotations enhance multi-modal retrieval**: We evaluate seven retrievers that integrate visual and textual cues to retrieve culturally relevant documents. We find fine-tuning retrievers on culture-targeted annotations leads to marked gains in retrieval accuracy, highlighting the value of explicit cultural supervision.
- **Benefits of culture-aware retrieval**: Culture-aware retrieval boosts task performance across VLMs, with lightweight models showing the better improvement. This suggests that such retrieval can seamlessly integrate into downstream VLM tasks, enhancing their performance.
- **Cross-cultural variation:** Evaluation across eight countries reveals that VLMs, both with and without cultural RAG exhibit distinct cultural preferences, with each model favoring different regional contexts, suggesting model-specific cultural biases.

## 2 RAVENEA CONSTRUCTION

We present RAVENEA, the first benchmark explicitly designed to assess how retrieval augmentation influences the visual cultural understanding capabilities of VLMs. It is derived originally from 30 countries; see Table 4. Each image is paired with 10 algorithm-ranked documents based on cultural relevance. To ensure high data quality while maximizing linguistic diversity across major language families, we sample data from eight countries spanning multiple continents: China, Nigeria, Russia, Spain, Mexico, India, Indonesia, and Korea.[2] We then apply human re-ranking–based annotation to the collected documents to construct RAVENEA. Consequently, RAVENEA comprises 1,868 instances, including 2,331 image-text questions, 655 image-caption pairs, and 18,680 image-document pairs, supporting two tasks: cVQA and cIC, respectively.

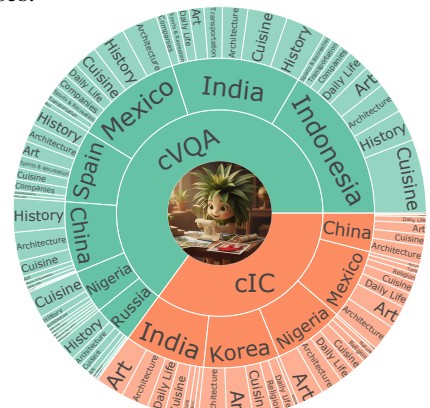

Figure 2: **RAVENEA**: A Multimodal **R**etrieval-**A**ugmented **V**isual cultur**E** u**N**d**E**rst**A**nding dataset. Geographic and categorical distribution of cultural references in RAVENEA.

### 2.1 TAXONOMY

Motivated by the increasing need for VLMs to understand cultural context (Yadav et al., 2025a), we categorize the cultural understanding capability into two core tasks and several detailed axes, as shown in Figure 2. Additional dataset statistics are provided in Appendix D.

**cVQA.** This task evaluates a model's ability to perceive and reason over culturally grounded visual content, often requiring external, culturally specific knowledge beyond the image itself. Building on the CVQA dataset (Romero et al., 2025), which contains culture-related QA pairs, we augment it with culturally relevant documents that supply the additional knowledge required for RAG evaluation. Therefore, each instance contains an image, a question, top 10 human-ranked documents based on culture relevance, and multiple-choice options, with only one correct answer.

**cIC.** This task assesses the VLM's ability to generate captions that are sensitive to and incorporate cultural nuances. It evaluates the model's generation sensitivity by requiring it to produce culturally contextualized captions for a given image, leveraging information from human-curated, culturally related documents. The task reuses cultural captions from the CCUB dataset (Liu et al., 2023) as

---

[2]List of languages by total number of speakers.

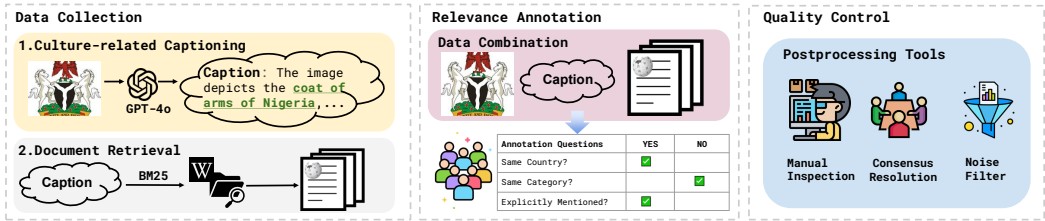

Figure 3: **RAVENEA construction pipeline. Left**: A two-stage retrieval process to match each image with relevant documents. **Middle**: Decomposition of cultural relevance into three interpretable dimensions to improve human annotation. **Right**: Postprocessing methods for quality control.

ground-truth references for this task. As a result, for cIC, each instance consists of an image, a human-written caption, and top 10 culturally-related documents.

## 2.2 DATASET CURATION

**Data collection.** As shown in Figure 3, the data collection process consists of two main steps: **(i) culture-related captioning** and **(ii) document retrieval**. For **culture-related captioning**, we generate culturally grounded captions for each image to facilitate more effective attachment of relevant documents. Since the CVQA lacks captions and the CCUB provides only brief descriptions, we employ GPT-4o[3] (Achiam et al., 2023) to generate richer, culturally informative captions (see the prompt example in Table 13).[4] For **document retrieval**, we begin with a coarse-grained filtering stage in which the generated cultural captions serve as queries to a BM25 retriever (Robertson & Zaragoza, 2009). This stage retrieves a pool of semantically relevant candidates from a large-scale corpus of over six million English Wikipedia articles.[5] To mitigate the impact of inaccurate captions and ensure precise document relevance, we then perform human annotation on the retrieved documents.

**Cultural relevance annotation.** Based on the initial BM25 retrieval results, we refine the cultural relevance label of retrieved documents via human annotation. For each image-caption pair, annotators are presented with the top 10 Wikipedia documents retrieved by BM25. They are asked to assess whether each document provides background or information that is relevant to the culture described in the caption or the image (see interfaces in Appendix L). Specifically, we decomposed cultural relevance into three interpretable and independently verifiable dimensions:

- **Country association**: *Is the topic of the Wikipedia article associated with the same country as the image and its caption?*
- **Topic alignment**: *Does the topic of the Wikipedia article align with the semantic category of the image and its caption?*
- **Explicit visual representation**: *Is the topic of the Wikipedia article explicitly mentioned or visually represented in the image and its caption?*

Each dimension is framed as a binary (True / False) question to reduce ambiguity and improve annotation consistency. However, for the **country association** dimension (the first listed), we introduce an additional label, *"Cannot be determined"*, to handle cases where this association is unclear from the annotator's perspective. Additionally, annotators are also instructed to include the title and URL of any relevant Wikipedia article they believe is missing from the top-10 retrieved results. These manually suggested articles are treated as the most culturally relevant references for the given image. Detailed annotation statistics, alongside positive, negative and missing ratios by countries, are presented in Appendix E.

**Quality control.** For caption quality, we employ two independent local annotators assessed both objective accuracy and subjective quality (naturalness and completeness) in a subset, achieving 92%

---

[3]Knowledge cutoff: Oct 01, 2023.

[4]We used LLMs to generate culture-informed captions solely as auxiliary references for annotators; these LLM outputs were not directly included in the final dataset.

[5]Wikimedia/wikipedia

and 94% accuracy with an inter-annotator agreement (IAA) Cohen's Kappa ($\kappa$) (Artstein, 2017) of 0.85, and yielding high subjective ratings (naturalness: 4.98/5, completeness: 4.66/5). The results demonstrate that the captions are reliable for downstream retrieval of Wikipedia documents. For annotation quality, a separate group of annotators were trained with detailed guidelines and a mock test before performing the actual annotation. A meta quality checker further validated a random subset, achieving a 98.2% acceptance rate. The IAA Cohen's Kappa ($\kappa$) between the meta checker and annotator on the sampled annotations is 0.83. More details are presented in Appendix F.

## 3 EXPERIMENTAL SETUP

With RAVENEA, we train and evaluate seven multimodal retrievers to retrieve culturally relevant Wikipedia documents using multimodal inputs, spanning both generative and discriminative paradigms. To enhance cultural awareness in the contrastive retrieval, we introduce Culture-Aware Contrastive (CAC) learning (details in Section 3.2), a supervised learning framework compatible with both CLIP (Radford et al., 2021) and SigLIP (Tschannen et al., 2025) architectures. Then, we evaluate the effectiveness of these retrievers with 17 widely used VLMs across the **cVQA** and **cIC** downstream tasks. More details are in Appendix I.

### 3.1 EVALUATION METRICS

For **multimodal retrieval**, the performance is evaluated using standard retrieval metrics, including Mean Reciprocal Rank (MRR) (Shi et al., 2012), Precision@k (P@k) (Järvelin & Kekäläinen, 2017), and Normalized Discounted Cumulative Gain (nDCG@k) (Wang et al., 2013), where $k \in \{1, 3, 5\}$.

For **cVQA**, we use accuracy as the evaluation metric, as all tasks follow a multiple-choice format. Accuracy is computed per VLM and per retriever, representing the proportion of correctly answered questions. For **cIC**, we employ several metrics for evaluating the generated caption, including ROUGE-L (Lin, 2004), CIDEr (Vedantam et al., 2015), BERTScore (Zhang* et al., 2020), and CLIPScore (Hessel et al., 2021).

Table 1: **Kendall's $\tau$ rank correlation (Kendall, 1938) between automatic metrics and human judgments for the cIC task.** Statistically significant correlations ($p < 0.05$) are marked with ✓. Our proposed metrics correlate stronger with human evaluation than the others.

| Rouge-L | CIDER | BERTScore | CLIPScore | RegionScore (ours) |
|---------|-------|-----------|-----------|--------------------|
| -0.172 ✗ | -0.316 ✗ | -0.011 ✗ | 0.139 ✗ | 0.442 ✓ |

To further evaluate the cultural relevance of the generated captions, we conducted a human study in which four researchers selected the most accurate captions from 14 VLMs.[6] We find a significant mismatch between automatic metrics and human judgments of cultural appropriateness (Table 1). To bridge this gap, we further introduce **RegionScore**, a novel evaluation metric designed to quantify the extent to which captions reference specific geopolitical regions. See details in Appendix G and H.

**RegionScore.** Let $\mathbb{I} = \{I_1, I_2, \ldots, I_n\}$ denote the set of images. For each image $I_i$, let $\boldsymbol{g}^{(i)}$ denote its generated caption, and let $\boldsymbol{c}^{(i)}$ represent the country name associated with $I_i$. We define $\mathbb{A}_i$ as the set of adjectives or demonyms corresponding to $\boldsymbol{c}^{(i)}$. For convenience, we introduce the set of region-related terms associated with $I_i$: $\mathbb{T}_{I_i} = \{\boldsymbol{c}^{(i)}\} \cup \mathbb{A}_i$. The instance-level $R(\boldsymbol{g}^{(i)}, I_i)$ metric for sample $I_i$ and $\boldsymbol{g}^{(i)}$ is defined as:

$$R(\boldsymbol{g}^{(i)}, I_i) = \begin{cases} 1 & \text{if } \exists\, w \in \boldsymbol{g}^{(i)} \text{ such that } w \in \mathbb{T}_{I_i}, \\ 0 & \text{otherwise.} \end{cases} \tag{1}$$

The overall RegionScore for predicted captions are then computed as:

$$RegionScore_{\mathbb{I}} = \frac{1}{|\mathbb{I}|} \sum_{I_i \in \mathbb{I}} R(\boldsymbol{g}^{(i)}, I_i), \quad RegionScore_{\mathbb{I}} \in [0, 1] \tag{2}$$

---

[6]The Qwen3-VL models were omitted from the human evaluation for the cIC task because the used model variants were not released at that time.

Table 2: **Performance with different retriever models.** Fine-tuned contrastive models consistently outperform their frozen counterparts (shown in gray). Ravenea-SigLIP and Ravenea-CLIP are SigLIP2-SO/14@384px and CLIP-L/14@224px models fine-tuned on RAVENEA, respectively.

| Method | MRR ↑ | P@1 ↑ | P@3 ↑ | P@5 ↑ | nDCG@1 ↑ | nDCG@3 ↑ | nDCG@5 ↑ |
|---|---|---|---|---|---|---|---|
| SigLIP2-SO/14@384px | 68.62 | 54.66 | 37.47 | 32.92 | 61.22 | 63.82 | 71.44 |
| CLIP-L/14@224px | 75.44 | 60.87 | 41.41 | 34.41 | 67.75 | 72.31 | 78.09 |
| VisualBERT | 61.33 | 46.71 | 33.14 | 29.42 | 55.76 | 59.42 | 65.92 |
| VL-T5 | 58.33 | 39.86 | 32.58 | 29.35 | 47.74 | 57.29 | 62.12 |
| LLaVA-OV-7B | 58.85 | 37.48 | 30.68 | 28.21 | 48.59 | 51.80 | 60.34 |
| **Ravenea-SigLIP (ours)** | 70.95 | 57.14 | 40.99 | 33.29 | 63.86 | 68.31 | 73.92 |
| **Ravenea-CLIP (ours)** | **82.17** | **72.05** | **45.76** | **36.77** | **77.08** | **78.96** | **84.09** |

RegionScore is designed to complement, rather than replace, existing general-purpose captioning metrics (e.g., CLIPScore), which primarily assess semantic alignment but do not explicitly capture geographic specificity. Therefore, RegionScore serves as a proxy for regional awareness by quantifying the proportion of captions that include explicit references of a country or its common demonyms, with higher values indicating stronger region awareness. In CCUB, RegionScore$_{\text{ground truth}}$ is 99%.

## 3.2 CULTURE-AWARE CONTRASTIVE LEARNING

Given an image $I_i$ associated with a set of textual documents $\mathbb{D} = \{D_i^1, D_i^2, \ldots, D_i^n\}$, each document $D_i^j$ is annotated with a binary label $y_i^j \in \{0, 1\}$, where $y_i^j = 1$ indicates cultural relevance and $y_i^j = 0$ indicates irrelevance. For each image-document pair $(I_i, D_i^j)$, we employ a shared vision-language encoder, such as CLIP, to obtain modality-specific representations: $\mathbf{E}_{I_i} = \mathcal{E}_V(I_i)$ for the visual input and $\mathbf{E}_{D_i^j} = \mathcal{E}_L(D_i^j)$ for the textual input. We then compute the cosine similarity score $s_i^t$ between $\mathbf{E}_{I_i}$ and each corresponding $\mathbf{E}_{D_i^j}$, resulting in a similarity vector $\mathbf{S}_i = [s_i^1, s_i^2, \ldots, s_i^n]$. Culture-awareness classification now amounts to:

$$\mathcal{L}_{\text{Culture Classify}} = -\frac{1}{|\mathbb{I}| \cdot |\mathbb{D}|} \sum_{I_i \in \mathbb{I}} \sum_{D_i^j \in \mathbb{D}} \left[ y_i^j \log \sigma(s_i^j) + (1 - y_i^j) \log(1 - \sigma(s_i^j)) \right], \quad (3)$$

where $\sigma(\cdot)$ denotes the sigmoid function.

To prioritize culturally relevant descriptions in the ranking, we apply a margin ranking loss between all pairs of descriptions with differing cultural relevance. For each image $I_i$, we compare all pairs $(D_i^j, D_i^k)$ such that $y_i^j = 1$ and $y_i^k = 0$, and encourage the model to assign a higher similarity score to the relevant description. The ranking loss is defined as:

$$\mathcal{L}_{\text{Rank}} = \frac{1}{|\mathbb{I}|} \sum_{i=1}^{|\mathbb{I}|} \sum_{\substack{j,k=1, \\ y_i^j=1, y_i^k=0}}^{|\mathbb{D}|} \max\left(0, \delta - (s_i^j - s_i^k)\right), \quad (4)$$

To mitigate the risk of overly similar positive text embeddings for the same image, we introduce a penalty that encourages intra-modal diversity among textual representations. We apply a diversity-promoting loss that forces the similarity between different text embeddings to be reduced while keeping each embedding highly similar to itself. Specifically, the penalty is formulated using an exponential function to emphasize the dissimilarity between embeddings:

$$\mathcal{L}_{\text{Diversity}}(\mathbf{S}_i) = -\sum_{t=1}^{|\mathbb{D}|} \log\left(\frac{\exp(s_i^t)}{\sum_{j=1}^{|\mathbb{D}|} \exp(s_i^j)}\right) \quad (5)$$

Then we can get the culture-aware contrastive loss:

$$\mathcal{L}_{\text{CAC}} = \frac{1}{3}\left(\mathcal{L}_{\text{Culture Classify}} + \mathcal{L}_{\text{Rank}} + \mathcal{L}_{\text{Diversity}}\right). \quad (6)$$

Table 3: **cVQA and cIC Performance w/ and w/o RAG.** Models in gray are frozen retrievers. The best results are in **bold**. VLMs augmented with finetuned retriever generally perform better.

| Retriever | Average | Open-source | | | | | | | | | | | | | | | | Proprietary |
|---|---|---|---|---|---|---|---|---|---|---|---|---|---|---|---|---|---|---|
| | | DeepSeek-VL2-Tiny | DeepSeek-VL2 | Qwen2.5-VL-3B | Qwen2.5-VL-7B | Qwen2.5-VL-72B | InternVL3-2B | InternVL3-8B | InternVL3-78B | Gemma3-4B | Gemma3-27B | Phi4-Multimodal | Pixtral-12B | LLaVA-OV-7B | Qwen3-VL-2B | Qwen3-VL-8B | Qwen3-VL-32B | GPT-4.1 |
| *cVQA_Accuracy↑* | | | | | | | | | | | | | | | | | | |
| W/O RAG | 67.1 | 49.8 | 68.9 | 62.7 | 52.6 | **83.3** | 64.1 | 71.3 | 84.2 | 66.5 | 79.4 | 42.6 | 73.2 | **62.7** | 37.8 | 73.7 | 81.3 | **86.6** |
| SigLIP2-SO/14@384px | 71.0 | 48.8 | 76.1 | 63.2 | 67.5 | 82.3 | 71.3 | 73.7 | **86.1** | 67.5 | 77.5 | 64.6 | 75.1 | 44.5 | 67.9 | 75.6 | 83.3 | 82.8 |
| CLIP-L/14@224px | 71.9 | 50.2 | **77.5** | 63.6 | 66.0 | 82.8 | 73.7 | 78.5 | 84.2 | 69.4 | 76.6 | 67.0 | **76.6** | 44.5 | 68.9 | 76.6 | 83.3 | 83.3 |
| **Finetuned Models** | | | | | | | | | | | | | | | | | | |
| VisualBERT | 67.6 | 48.8 | 71.8 | 61.7 | 64.1 | 78.5 | 66.5 | 67.5 | 82.3 | 63.6 | 75.1 | 61.2 | 68.4 | 37.3 | 60.8 | 73.2 | 83.7 | 85.2 |
| VL-T5 | 66.1 | 46.6 | 72.7 | 58.4 | 65.1 | 78.5 | 65.1 | 67.9 | 80.4 | 58.4 | 75.6 | 58.9 | 62.7 | 35.4 | 60.3 | 75.6 | 78.9 | 83.7 |
| LLaVA-OV-7B | 67.9 | 50.1 | 66.0 | 60.3 | 67.0 | 81.8 | 66.5 | 66.0 | 84.2 | 65.1 | 75.1 | 60.8 | 68.9 | 37.3 | 59.8 | 77.0 | 83.7 | 84.7 |
| **Ravenea-SigLIP (ours)** | 72.6 | 48.8 | 76.1 | **68.4** | 67.9 | 81.3 | 73.2 | 75.6 | 85.6 | 70.3 | 79.4 | 69.4 | 75.6 | 47.8 | 68.9 | 77.0 | **84.2** | 85.6 |
| **Ravenea-CLIP (ours)** | 73.3 | **50.7** | 75.1 | 64.1 | **69.9** | 82.3 | **76.1** | **81.3** | 85.2 | 69.9 | 79.9 | 69.4 | 75.1 | 50.2 | **69.4** | 77.5 | **84.2** | 86.1 |
| *cIC_RegionScore↑* | | | | | | | | | | | | | | | | | | |
| W/O RAG | 57.5 | 26.5 | 63.3 | 53.1 | 69.4 | 71.4 | 38.8 | 49.0 | 65.3 | 73.5 | **75.5** | 28.6 | 53.1 | 49.0 | 59.2 | 61.2 | **73.5** | 67.3 |
| SigLIP2-SO/14@384px | 54.8 | 30.6 | 59.2 | 51.0 | 63.3 | 65.3 | 44.9 | 53.1 | 57.1 | 65.3 | 69.4 | 46.9 | 53.1 | 42.9 | 49.0 | 59.2 | 59.2 | 61.2 |
| CLIP-L/14@224px | 65.1 | 38.8 | 69.4 | 59.2 | 71.4 | 69.4 | 65.3 | 65.3 | 71.4 | 73.5 | 73.5 | 61.2 | 69.4 | 51.0 | 63.3 | 67.3 | 71.4 | 65.3 |
| **Finetuned Models** | | | | | | | | | | | | | | | | | | |
| VisualBERT | 57.4 | 36.7 | 57.1 | 53.1 | 61.2 | 69.4 | 44.9 | 59.2 | 61.2 | 63.3 | 69.4 | 42.9 | 57.1 | 44.9 | 55.1 | 65.3 | 69.4 | 65.3 |
| VL-T5 | 57.0 | 34.7 | 61.2 | 55.1 | 65.3 | 73.5 | 46.9 | 53.1 | 61.2 | 61.2 | 67.3 | 44.9 | 59.2 | 38.8 | 53.1 | 63.3 | 67.3 | 63.3 |
| LLaVA-OV-7B | 56.8 | 38.8 | 63.3 | 51.0 | 63.3 | 69.4 | 44.9 | 57.1 | 59.2 | 65.3 | 71.4 | 40.8 | 55.1 | 46.9 | 49.0 | 61.2 | 67.3 | 61.2 |
| **Ravenea-SigLIP (ours)** | 60.6 | 30.6 | **73.5** | 57.1 | 67.3 | 71.4 | 44.9 | 59.2 | 69.4 | 67.3 | 69.4 | 57.1 | 61.2 | 40.8 | 53.1 | 65.3 | 71.4 | **71.4** |
| **Ravenea-CLIP (ours)** | 68.8 | 46.9 | **73.5** | 61.2 | **77.6** | 75.5 | 69.4 | 67.3 | 73.5 | 75.5 | 75.5 | 63.3 | 71.4 | 55.1 | 67.3 | 73.5 | 73.5 | 69.4 |

## 4 EXPERIMENTAL RESULTS

### 4.1 MAIN RESULTS

**Multimodal retrieval results.** We perform a comprehensive evaluation of both frozen and fine-tuned retrievers, and present the results in Table 2. We find that fine-tuned models, particularly those based on contrastive learning, consistently outperform their frozen counterparts. For instance, Ravenea-CLIP achieves a substantial improvement in P@1, rising from 60.87% to 72.05%, and sets a new SOTA across all evaluation metrics. Although SigLIP2-SO/14@384px (Tschannen et al., 2025) also benefits from fine-tuning, the performance gains are comparatively modest. In contrast, models such as LLaVA-OV-7B (Li et al., 2025), VL-T5 (Cho et al., 2021), and VisualBERT (Li et al., 2019; 2020) lag behind after fine-tuning, even underperforming relative to frozen baselines. This underperformance likely stems from the fact that models such as LLaVA-OV-7B and VisualBERT were originally pretrained for generative tasks with different objectives, whereas CLIP-L/14@224px (Radford et al., 2021) and SigLIP2-SO/14@384px were explicitly trained for similarity-based alignment, providing them with a structural advantage in retrieval settings.

**Downstream tasks.** We present downstream performance on both cVQA and cIC tasks in Table 3. The results demonstrate the efficacy of incorporating culture-aware retrieval augmentation. Our proposed fine-tuned retrievers, Ravenea-SigLIP and Ravenea-CLIP, consistently achieve the highest scores across both cVQA and cIC tasks, frequently outperforming both the non-RAG baselines and frozen retrievers (SigLIP2 / CLIP). Specifically, averaged over all VLMs, Ravenea-CLIP achieves the highest performance for both tasks, improving +6.2% in cVQA accuracy (67.1% → 73.3%), and +11.3% in cIC RegionScore (57.5% → 68.8%). While CLIP-L/14@224px also offers improvements, culture-aware contrastive learning consistently unlocks further potential. Furthermore, without retrieval, larger VLMs still achieve strong absolute performance on both tasks (e.g., InternVL3-78B at 84.2% on cVQA, and Gemma3-27B at 75.5% on cIC). Retrieval contents, however, disproportionately benefits small and mid size models. For lightweight models (≤ 8B parameters), Ravenea-CLIP narrows the gap to the larger models. For example, Qwen3-VL-8B improves to 77.5% on cVQA, approaching the performance of the models an order of magnitude larger, highlighting the potential of culture-aware retrieval.

## 4.2 FURTHER FINDINGS

Culture-aware retrieval augmentation substantially benefits VLMs across both cVQA and cIC tasks, compared to their non-RAG counterparts. In this section, we explore the margin of improvement, cultural preference, the effectiveness of cultural annotation, and the impact of context length.

> • **Finding 1.** *While larger VLMs see limited gains from different retrievers, lightweight VLMs benefit substantially from culturally grounded retrieval contexts.*

To evaluate the scaling behavior of VLMs incorporating with `Ravenea-CLIP`, we select four model families spanning from 2B to 78B parameters and quantify the performance improvement $\Delta$ relative to non-RAG baselines. As shown in Figure 4, the impact of `Ravenea-CLIP` does not scale monotonically with model size: smaller models exhibit the most substantial gains, while larger models (27B to 78B) exhibit only modest improvements. For instance, Qwen3-VL-2B achieves +31.6% absolute improvement on cVQA and +8.1% on cIC, whereas Qwen3-VL-32B shows only +2.9% better on cVQA and no gain on cIC. This trend indicates that larger models may already internalize much of the relevant knowledge, especially the Wikipedia content retrieved by `Ravenea-CLIP`, resulting in redundancy and a plateau in performance. Overall, these results suggest that `Ravenea-CLIP` serves as an

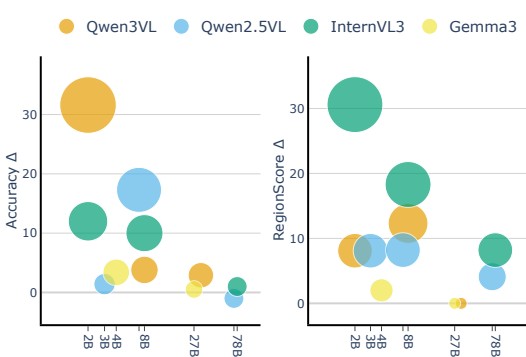

Figure 4: **Performance improvements per model family with `Ravenea-CLIP`.** Scaling VLMs yields marginal gains on both cVQA and cIC tasks. "$\Delta$" represents the change with `Ravenea-CLIP` relative to the non-RAG baseline.

effective plug-and-play enhancement for lightweight models on both cVQA and cIC tasks, providing diminishing yet consistent gains as model size increases.

> • **Finding 2.** *With RAVENEA, VLMs show cultural disparities with and without RAG. Moreover, large inter-model variance highlights model-specific biases.*

For geographic variability, we evaluate all models using `Ravenea-CLIP` across a range of countries for both tasks, as shown in Figure 5. For cVQA, most VLMs exhibit substantially diminished performance on culture-specific questions regarding Nigeria and Indonesia, in contrast to their performance on questions under other national contexts. Interestingly, questions related to Spanish culture reveal high inter-model variance, with accuracy differentials reaching up to 50%, underscoring significant discrepancies in cultural representation across models. For cIC, VLMs consistently underperform on images and documents associated with Mexico cultural contexts, while achieving the highest RegionScores on Indian culture-related inputs. Model performance on Mexico culture

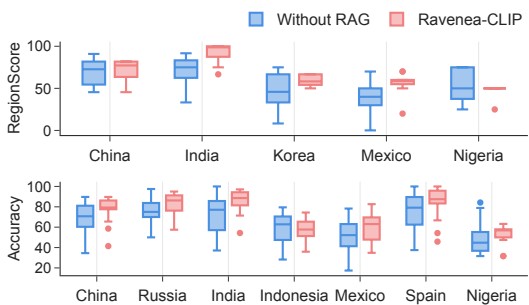

Figure 5: **Geographic variability in VLM performance.** While `Ravenea-CLIP` yields gains in general, there's still variance in both baseline and relative improvement across geographic regions.

is particularly volatile, indicating inconsistent cultural grounding across architectures. By comparison, Korean and Chinese cultural inputs yield more stable performance across models, suggesting entrenched model-specific preferences in cultural alignment (see more results in Appendix J).

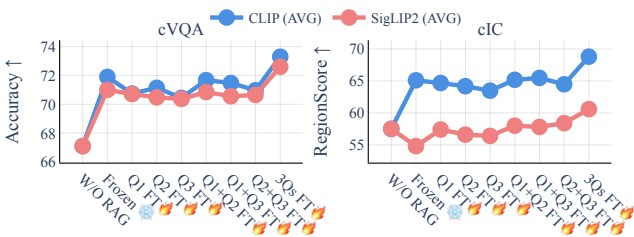 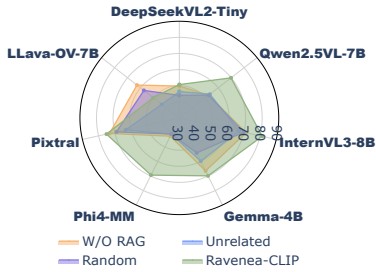

(a) Finetuning on three questions consistently achieves the better performance (averages shown across 16 open-source VLMs). ❄ denotes frozen models, FT 🔥 indicates finetuning on the corresponding annotation questions.

(b) Ravenea-CLIP generally outperform the other retrieval methods on cVQA task.

Figure 6: **Ablations for annotations and retrieval content.** Combining all three culture-relevant annotation questions consistently achieves the highest performance on both downstream tasks, validating the effectiveness of RAVENEA setup.

> • **Finding 3.** *Generally, models benefit most when trained with the full set of culture-relevant annotations. Culturally grounded retrieval substantially outperforms random or irrelevant contexts, underscoring both the necessity of culture-aware contexts. Furthermore, more or longer retrieved context length yield diminishing gains.*

**Annotation questions ablations.** We further perform ablation studies across diverse combinations of annotation questions to assess their impact on downstream performance. Specifically, we evaluate 16 open-weight VLMs equipped with either SigLIP2-SO/14@384px or CLIP-L/14@224px, each fine-tuned on datasets constructed using varying subsets of culture-relevant annotations. From Figure 6a, we can observe that leveraging all three questions (Q1 regarding country association; Q2 for topic alignment; Q3 for explicit visual representation) yields the strongest performance on both cVQA and cIC tasks. For the cVQA task, we find Q1 provides the most significant benefit to Ravenea-SigLIP, whereas Ravenea-CLIP gains more from Q2. Among all pairwise combinations, the joint supervision from Q1 and Q2 proves slightly more effective than other pairs. In the cIC task, both Ravenea-SigLIP and Ravenea-CLIP achieve better performance improvements when trained with data derived from Q1, compared to other single-question sets. For pairwise combinations, Ravenea-CLIP benefits most from the Q1+Q3 combination, while Ravenea-SigLIP shows a clear preference for the Q2+Q3 setup.

**Context relevance ablations.** Recent work (Roth et al., 2023; Ma et al., 2025) suggests that randomly augmented prompts may yield performance gains even in the absence of meaningful knowledge, raising concerns that improvements attributed to retrieval could in fact stem from other factors, such as superficial prompt lengthening and so on. To rigorously assess the effectiveness of culturally grounded retrieval, we design two control conditions: *random character sequences*, *culturally unrelated documents*. For *random character sequences*, we randomly generate synthetic strings in which each string consists of 100 to 300 substrings, and each substring contains 4 to 8 randomly sampled alphabetic characters, ensuring comparable token budgets without introducing interpretable content. For *culturally unrelated documents*, we randomly sample hundreds of Wikipedia articles from countries not represented in RAVENEA, thereby maintaining language structure and topical coherence while explicitly breaking cultural relevance. Together, these conditions isolate whether gains arise from the related cultural context or merely from the presence of additional text. As shown in Figure 6b, models augmented with Ravenea-CLIP consistently outperform other configurations, highlighting both the effectiveness of cultural retrieval and the robustness of the RAVENEA evaluation protocol. More results are provided in Appendix J Table 11.

**Context length ablations.** We also extend our study with a controlled analysis of retrieval length, comparing three additional context granularities: *paragraph-level*, *full top-1 document*, and *full top-3 documents*. As shown in Figure 7, providing a compact, high-salience context (*Top-1 256 tokens*) consistently achieves the best or tied-best performance across most model scales. In contrast, expanding the context to full documents or multiple documents often leads to marginal gains or

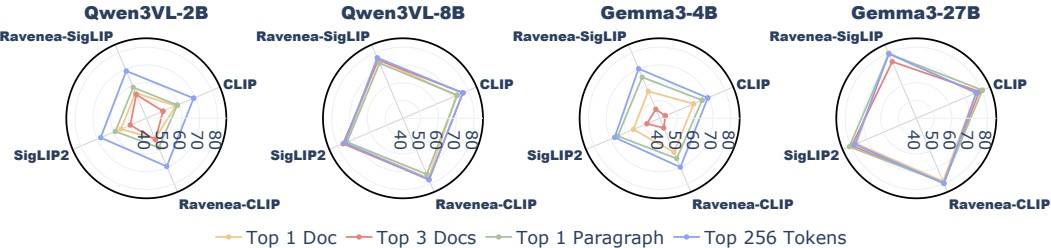

Figure 7: **Context length ablations for cVQA task.** With `Ravenea-CLIP`, adding longer or multiple documents yields diminishing returns, highlighting the importance of the context length.

degradation. This trend is particularly evident for lightweight models, which appear more sensitive to the context length. Even for larger models, additional context yields only marginal improvements despite substantially increasing input length. These results indicate that simply retrieving more content does not necessarily translate into better performance, especially for the lightweight models.

## 5 RELATED WORK

**Retrieval augmentation for cultural understanding.** Retrieval augmentation has demonstrated significant efficacy for culture-related NLP tasks (Conia et al., 2024; Hu et al., 2024; Fung et al., 2023). Some works have explored retrieval-augmented approaches to cultural question answering. One line of work implemented retrieval using the World Values Survey to enable in-context learning for culturally contextualized questions (Seo et al., 2025; Yadav et al., 2025b), while others enhanced cultural question answering by retrieving information from web searches and knowledge bases (Lertvittayakumjorn et al., 2025). In cross-cultural applications, retrieval has also been shown effective for adapting recipes across cultural contexts (Hu et al., 2024). For multimodal understanding, however, the direction of culture-relevant retrieval remains underexplored.

**Vision-language culture datasets.** Recent studies examine cultural understanding in VLMs through multicultural VQA (Liu et al., 2021; Nayak et al., 2024; Liu et al., 2025). Some researchers focus on culturally grounded visual categories such as cuisine, where models must distinguish dishes that are visually similar yet culturally distinct, highlighting fine-grained, geographically situated knowledge gaps (Winata et al., 2025; Li et al., 2024). Socioeconomic signals have also been explored, for example by studying how models infer or misinfer income level and living standards from visual cues (Nwatu et al., 2025). Others benchmark cultural entity recognition using Wikipedia-based prompts (Nikandrou et al., 2025), curate culture-specific image sets for value-based tasks (Yadav et al., 2025b) or concept-based image retrieval without explicitly considering cultural relevance (Bhatia et al., 2024). While prior efforts highlight the limitations of current VLM, they mainly assess the performance on retrieval and downstream tasks respectively, which are not suitable for probing how RAG shapes visual-cultural understanding. Our work specifically aims to fill this research gap.

## 6 CONCLUSION

We presented RAVENEA, a new benchmark for multimodal retrieval-augmented visual culture understanding that enables the evaluation of how external cultural knowledge improves VLMs. By pairing culture-centric visual question answering and culture-informed captioning with 11,396 human-ranked Wikipedia documents across eight countries and eleven categories, RAVENEA provides the first large-scale testbed for studying multimodal retrieval in culturally grounded settings. Our experiments across seven multimodal retrievers and seventeen VLMs show that incorporating cultural grounding consistently improves both retrieval quality and downstream task performance. Notably, lightweight VLMs benefit significantly from culture-aware retrieval, highlighting multimodal retrieval augmentation as an efficient way to enhance culturally situated understanding and reasoning. Overall, RAVENEA offers a standardized framework for diagnosing limitations and advancing research in multimodal retrieval, grounding, and generation for VLMs in culture domain. We hope this benchmark will drive future work toward more culture-sensitive and globally inclusive multimodal intelligence.

## ETHICS STATEMENT

This work focuses on improving the cultural awareness of VLMs through retrieval-augmented methods. All data used in the construction of the RAVENEA benchmark were sourced from publicly available datasets and Wikipedia, a community-curated open-access knowledge base. To protect individual privacy, we apply automated face detection and blur all identifiable faces in images prior to release. To mitigate cultural bias and ensure broad representation, the benchmark includes images and documents spanning eight countries and eleven cultural domains, curated and annotated by a diverse group of annotators. We provide detailed documentation of the annotation process and guidelines to support transparency and reproducibility. While enhancing cultural understanding is a central goal of this work, we acknowledge that culture is inherently complex, dynamic, and context-dependent. Consequently, the benchmark cannot capture the full richness of any cultural context.

Unavoidably, when models fail, errors may go beyond reduced accuracy, leading to misattributed traditions or distorted cultural practices. In heritage contexts, such failures may undermine trust in digital tools or inadvertently disrespect the communities represented. Reliance on broad but uneven sources like Wikipedia can further compound these risks by amplifying existing biases and privileging cultures with richer digital footprints, echoing broader challenges in AI for cultural heritage (Münster et al., 2024). Such distortions could affect heritage management by marginalizing underrepresented voices, eroding local agency, or encouraging overreliance on automated outputs. **To mitigate these risks, we emphasize safeguards such as human-in-the-loop validation with cultural experts, transparent documentation of dataset limitations, and fairness-oriented evaluation protocols**. Looking forward, retrieval grounded in domain-specific knowledge bases curated by GLAM (Galleries, Libraries, Archives, and Museums) institutions offers a path toward higher-quality cultural data. We also highlight the importance of systematically incorporating expert human evaluation to validate both cultural relevance and interpretive accuracy.

Finally, this work does not involve any personally identifiable information (PII), biometric data, or sensitive attributes. All human annotators were compensated fairly, and data annotation adhered to ethical guidelines for responsible research.

## ACKNOWLEDGMENT

We would like to thank all the annotators for their work. Special thanks to Nico Lang, Peter Ebert Christensen, Zhaochong An, Stella Frank, Srishti Yadav, for providing helpful research advice. This project is mainly supported by the National Research Foundation Singapore under the AI Singapore Programme (AISG Award No: AISG3-RPGV-2025-016). Jiaang Li and Serge Belongie are supported by the Pioneer Centre for AI, DNRF grant number P1. Ivan Vulić is supported by a personal Royal Society University Research Fellowship *'Inclusive and Sustainable Language Technology for a Truly Multilingual World'* (no 221137). Wenyan Li is supported by the Lundbeck Foundation (BrainDrugs grant: R279-2018-1145). Yang Deng is support by the Lee Kong Chian Fellowship awarded by Singapore Management University.

## REPRODUCIBILITY STATEMENT

Section 2 and Appendix E, F and L details the data collection and processing steps. Experimental setups are provided in Appendix I. Code and dataset are available here.

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

## A    THE USAGE OF LARGE LANGUAGE MODELS (LLMS)

We use LLMs as writing aids, limited to polishing grammar, improving clarity, and refining readability. Moreover, during data curation, we used LLMs to generate culture-informed captions solely as auxiliary references for annotators; these LLM outputs were not directly included in the final dataset. All technical content, results, and conclusions are original and authored entirely by the listed contributors. The responsibility for the manuscript's scientific validity and integrity rests fully with the authors.

## B    LIMITATIONS

While RAVENEA establishes a solid foundation for advancing the study of visual culture understanding with retrieval augmentation, it has three limitations that warrant future attention. First, due to budgetary constraints, the dataset's cultural scope is currently limited to eight countries and eleven categories. Although this selection introduces meaningful diversity, it does not comprehensively represent the global spectrum of cultural perspectives, particularly those of underrepresented or marginalized communities. Second, our use of Wikipedia as the primary external knowledge source introduces inherent biases and may lack the depth, plurality, and contextual richness necessary for nuanced cultural interpretation. Finally, due to resource limitations, we were unable to include certain proprietary VLMs that require paid APIs, such as Gemini 2.5 Pro (Team et al., 2023) and Claude Opus 3.7 (Anthropic, 2024). We hypothesize that their performance would be comparable to GPT-4.1 (Achiam et al., 2023), which was included in our evaluation, but this remains an open empirical question.

## C    FUTURE DIRECTIONS

Our work opens several avenues for advancing visual culture understanding in multimodal models. In constructing RAVENEA, we algorithmically generated a raw dataset spanning 30 countries, comprising over 3,900 images and 78,000 documents (Table 4). Due to budget constraints, annotations were limited to data from eight countries. Therefore, extending RAVENEA to include more countries, cultural categories, and diverse knowledge sources, beyond Wikipedia, would improve coverage and reduce institutional bias. Second, future benchmarks could include richer tasks beyond cVQA and cIC, such as culture-grounded object recognition, historical retrieval, and symbolic interpretation, to better capture cultural semantics. Third, our results suggest a need for culturally-aware evaluation metrics, particularly for text generation. The limited effectiveness of retrieval augmentation in larger models also warrants further study, especially regarding how cultural knowledge is integrated and utilized. Exploring retrieval pipelines grounded in domain-specific knowledge bases curated by GLAM (Galleries, Libraries, Archives, and Museums) institutions could provide higher-quality cultural data and more reliable grounding. Finally, systematic human evaluation by cultural experts remains indispensable for validating both relevance and interpretive accuracy. Together, these directions aim to support the development of more culturally-sensitive and globally robust vision-language models.

## D    DATA STATISTICS

Table 4: **Statistics of the RAVENEA dataset.** The dataset is constructed by curating existing sources and augmenting them with wiki-derived documents to broaden cultural knowledge coverage and enhance content diversity.

| Dataset | Countries | Images | Documents | Pairs | Questions | Captions |
|---|---|---|---|---|---|---|
| CVQA$_{selected}$ | 7 | 1,213 | 8,194 | 12,130 | 2,331 | - |
| CCUB$_{selected}$ | 5 | 655 | 4,354 | 6,550 | - | 655 |
| RAVENEA | 8 | 1,868 | 11,396 | 18,680 | 2,331 | 655 |
| BM25 filtered | 30 | 3,912 | 78,240 | - | - | - |

We then apply cryptographic hashing (SHA-256) to identify and remove duplicate images, resulting in a cleaner and more distinct cultural image set. Consequently, RAVENEA comprises images collected

Table 5: Statistics of images in each country in RAVENEA.

| India | Mexico | Indonesia | Nigeria | China | Korea | Spain | Russia |
|-------|--------|-----------|---------|-------|-------|-------|--------|
| 408   | 347    | 309       | 223     | 214   | 148   | 142   | 77     |

Table 6: Statistics of images in each category in RAVENEA.

| Architecture | Cuisine | Daily Life | History | Art | Companies | Sports & Recreation | Transportation | Religion | Nature | Tools |
|--------------|---------|------------|---------|-----|-----------|---------------------|----------------|----------|--------|-------|
| 403          | 402     | 347        | 278     | 86  | 80        | 73                  | 68             | 52       | 31     | 21    |

from eight countries across four continents, spanning eleven distinct categories. The distribution of images by country and by category is detailed in Tables 5 and 6, respectively. During constructing RAVENEA, we have constructed a comprehensive, algorithmically generated raw dataset covering 30 countries, comprising over 3,900 images and 78,000 documents (see Table 4). Owing to budgetary limitations, for RAVENEA, we have only annotated data from eight of these countries. Overall, we believe that both the raw algorithmically generated data and our human-labeled RAVENEA could be a valuable source for improving the cultural understanding in VLM tasks.

# E  ANNOTATION DETAILS

Based on the initial BM25 retrieval results, we refine the cultural relevance label of retrieved documents via human annotation. We found that directly asking annotators to rate overall cultural relevance on a continuous scale (e.g., 0–10) led to unreliable and inconsistent labels. This difficulty arises from several factors: (1) the semantic meaning of intermediate scores is ambiguous, (2) annotators tended to overemphasize a few salient visual elements (Chen et al., 2015), and (3) small numerical differences (e.g., between 5 and 6) often fail to reflect meaningful distinctions, especially given the cognitive load of processing lengthy Wikipedia documents, resulting in intra-annotator variance even on repeated examples. Instead, given an image–caption–document triplet, we decomposed cultural relevance into three interpretable and independently verifiable dimensions: **country association**, **topic alignment**, and **explicit visual representation**.

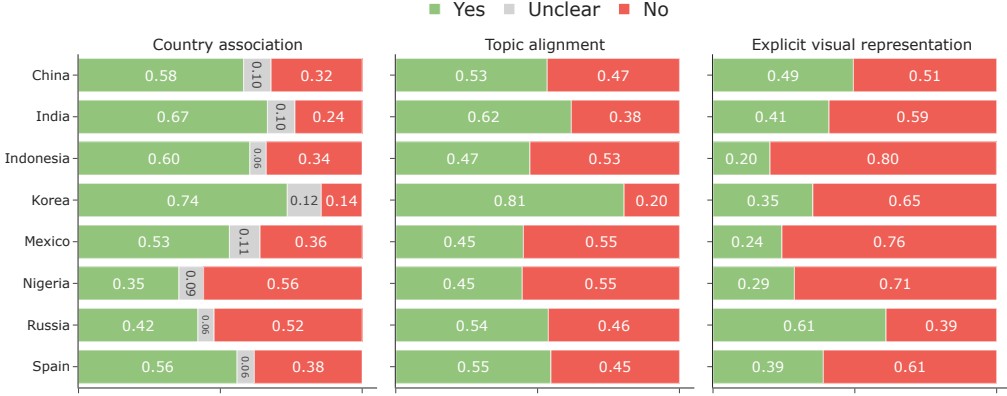

Figure 8: Statistics on the Yes / No response distributions for the annotation questions.

Prior to the annotation process, all annotators are required to carefully review a detailed instruction file outlining the relevance criteria and annotation guidelines. To ensure proper understanding of the guidelines, annotators are required to complete a mock annotation test and correctly answer all questions before proceeding with the actual annotation tasks. For each image-caption pair, annotators are presented with the top 10 Wikipedia documents retrieved by BM25. They are asked to assess whether each article provides meaningful background or contextual information that is directly

relevant to the cultural elements described in the caption or depicted in the image. The statistics of the annotations are presented in Figure 8.

Table 7: Statistics of missing documents, along with the positive and negative ratios for each country.

| Country | Positive & Negative Ratio(%) | Missing Ratio(%) |
|---|---|---|
| India | 54.42 / 45.58 | 9.56 |
| Russia | 54.42 / 45.58 | 0.0 |
| Spain | 48.17 / 51.83 | 0.0 |
| Nigeria | 34.47 / 65.53 | 0.45 |
| Mexico | 38.37 / 61.63 | 25.55 |
| Indonesia | 41.26 / 58.74 | 0.32 |
| Korea | 69.59 / 30.41 | 19.59 |
| China | 56.12 / 43.88 | 10.75 |

As shown in Table 7, regions such as Mexico (25.55%) and Korea (19.59%) exhibit notably high missing ratios, underscoring the indispensable role of human expertise in identifying culturally specific documents that purely automated retrieval methods fail to capture. This missing ratio serves as an informative diagnostic metric: higher values directly reflect the effectiveness of our human-in-the-loop pipeline in enriching the dataset with highly local and otherwise overlooked content, ultimately attesting to the superior cultural coverage and quality of RAVENEA. Furthermore, Table 7 reports the distribution of positive versus negative labels for document relevance across regions, offering additional insight into cultural representation variance.

## F  QUALITY CONTROL

To evaluate the quality of the GPT-4o-generated captions, we randomly sampled 50 captions from the subset of Chinese images, and employ two local checkers to check the quality in Table 8. Each annotator assessed the accuracy of the captions, and we report their individual accuracies along with the IAA to measure consistency. Beyond objective correctness, we also asked annotators to provide subjective ratings (5-point Likert scale) of caption quality along two dimensions: naturalness (fluency and readability) and completeness (coverage of visual content). The results further show the GPT-4o caption is suitable for the next step to retrieve the Wikipedia documents.

Table 8: Generated caption quality check. IAA denotes inter-annotator agreement.

| Acc. (Annotator 1) | Acc. (Annotator 2) | IAA | Naturalness (1–5) | Completeness (1–5) |
|---|---|---|---|---|
| 92% | 94% | 0.85 | 4.98 | 4.66 |

To ensure the quality and consistency of our annotations, we implement several quality control methods. First of all, prior to the annotation process, all annotators are required to carefully review a detailed instruction file outlining the relevance criteria and annotation guidelines. To ensure proper understanding of the guidelines, annotators are required to complete a mock annotation test and correctly answer all questions before proceeding with the actual annotation tasks (see Figure 13). We also perform an additional quality check on a subset of the dataset. Specifically, for each selected countries, we employ an additional local quality checker who is tasked with manually reviewing the annotations to verify their accuracy and adherence to the guidelines. The quality checker reviews a random sample of annotated items, focusing on both the relevance labels and the justification behind any edge cases, such as borderline relevance or use of the "Cannot be determined" label. If inconsistencies or deviations from the annotation guidelines are identified, the affected samples are flagged for re-annotation. The overall acceptance rate from the meta quality checkers is 98.2%. The inter-annotator agreement (IAA) Cohen's Kappa ($\kappa$) between the meta checker and annotator on the sampled annotations is 0.83.

## G    CORRELATION BETWEEN AUTOMATIC METRICS AND HUMAN JUDGMENTS

To assess the correlation between human preferences and automatic evaluation metrics, we compute Kendall's $\tau$ rank correlation. Human annotations are segmented according to the output chunks produced by corresponding 14 VLMs (no Qwen3VL family) with each retriever variant (`Ravenea-CLIP`-based, CLIP-based, and non-RAG). Within each segment, we calculate the selection winning ratio for each retrieval method, yielding a human preference vector formed by concatenating these ratios across all evaluation instances. For the automatic evaluation, we extract BERTScore, CIDEr, ROUGE-L, and CLIPScore for each corresponding retrieval variant and VLM. These scores are similarly concatenated into a metric-based vector. Finally, we compute Kendall's $\tau$ between the human and metric vectors to quantify the consistency between automatic rankings and human judgments.

## H    DETAILS OF HUMAN EVALUATION IN THE cIC TASK

We randomly sample 10 images and generate captions using 14 VLMs under three configurations: `Ravenea-CLIP`, CLIP, and a no-retrieval baseline, yielding 420 captions in total.[7] Four expert annotators participated in the evaluation. For each image, they were presented with caption triplets—one per retrieval configuration—and asked to assess approximately 35 such triplets each. Annotators are instructed to select the caption that most accurately and appropriately reflects the cultural context depicted. To evaluate annotation consistency, we randomly sample 30 triplets and assign them to a fifth expert annotator. Inter-annotator agreement (IAA) between this annotator and the original four, measured using Cohen's $\kappa$, is 0.595.

## I    IMPLEMENTATION DETAILS

### I.1    DATASET.

We integrate responses from three annotation questions per data point into a continuous scale ranging from $-3$ to $3$, where higher values indicate stronger cultural relevance. To ensure fair evaluation across regions, we adopt the train-validation-test split strategy from class-imbalance loss (Cui et al., 2019), yielding an approximate 85-5-10 data split. We also performed strict de-duplication on the training and validation sets, guaranteeing that no identical documents appear across the training, validation, testing splits. This setup guarantees that validation and test sets contain a balanced set of images per country in per task (Figure 9), thereby mitigating evaluation bias from skewed geographic distributions. During training, we

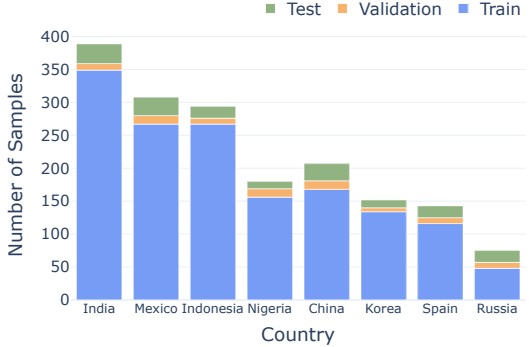

Figure 9: Data distributions across eight countries. Restrict the dataset to the images paired with at least one document annotated by human raters as culturally relevant.

employ random cropping for images as a sample-level augmentation to normalize the distribution of training instances across countries, further mitigating region-specific sampling biases.

### I.2    MULTIMODAL RETRIEVERS.

We finetune CLIP-L/14@224px (Radford et al., 2021), SigLIP2-SO/14@384px (Tschannen et al., 2025). We also finetune a VisualBERT-based (Li et al., 2019; 2020) reranker following standard BERT-style setups (Nogueira & Cho, 2019), and adapt two multimodal generative models, such as VL-T5 (Cho et al., 2021) and LLaVA-OneVision-7B (Li et al., 2025), for end-to-end document retrieval (Chen et al., 2022; Yuan et al., 2024; Sun et al., 2023). We integrate responses from three annotation questions per data point into a continuous scale ranging from $-3$ to $3$, where higher values indicate stronger cultural relevance.

---

[7]The variants of the Qwen3-VL models we used were not released at the time of this evaluation.

Table 9: Hyperparameters for finetuning on five models.

| Hyperparameters | VisualBERT | VL-T5 | LLaVA-OV-7B | CLIP-L/14@224px | SigLIP2-SO/14@384px |
|---|---|---|---|---|---|
| batch size | 32 | 128 | 4 | 64 | 64 |
| lr | 2e-4 | 1e-4 | 1e-4 | 1e-5 | 1e-5 |
| lr warmup ratio | - | 0.1 | - | 0.03 | 0.03 |
| Max Epoch | 100 | 20 | 10 | 2 | 2 |
| Patience | 10 | 10 | 3 | - | - |
| early stopping | Yes | Yes | Yes | No | No |
| optimizer | | | AdamW | | |
| Using LORA | No | No | Yes | No | No |

**Hyperparameters.** In this work, we adopt different sets of hyperparameters as VisualBERT, VL-T5, LLaVA-OV-7B, CLIP-L/14@224px, SigLIP2-SO/14@384px. For VisualBERT, VL-T5, and LLaVA-OV-7B we follow the setting in (Chen et al., 2022; Yuan et al., 2024; Sun et al., 2023). We show the training hyperparameters in rerankering experiments for all models in Table 9. All experiments are conducted using a maximum of 2 Nvidia H100 GPUs.

### I.3 VISION-LANGUAGE MODELS

Table 10: Model details: Hugging Face model names.

| Model | Hugging Face Model Name |
|---|---|
| LLaVA-OneVision-7B (Li et al., 2025) | `llava-hf/llava-onevision-qwen2-7b-ov-hf` |
| Phi-4-Multimodal (Abouelenin et al., 2025) | `microsoft/Phi-4-multimodal-instruct` |
| Pixtral (Agrawal et al., 2024) | `mistral-community/pixtral-12b` |
| Qwen2.5VL family (Bai et al., 2025b) | `Qwen/Qwen2.5-VL-3B-Instruct` |
| | `Qwen/Qwen2.5-VL-7B-Instruct` |
| | `Qwen/Qwen2.5-VL-72B-Instruct-AWQ` |
| Qwen3VL family (Bai et al., 2025a) | `Qwen/Qwen3-VL-2B-Instruct` |
| | `Qwen/Qwen3-VL-8B-Instruct` |
| | `Qwen/Qwen3-VL-32B-Instruct-FP8` |
| DeepSeek-VL2 family (Wu et al., 2024) | `deepseek-ai/deepseek-vl2` |
| | `deepseek-ai/deepseek-vl2-tiny` |
| InternVL3 family (Zhu et al., 2025) | `OpenGVLab/InternVL3-2B` |
| | `OpenGVLab/InternVL3-8B` |
| | `OpenGVLab/InternVL3-78B-AWQ` |
| Gemma3 family (Team et al., 2025) | `google/gemma-3-4b-it` |
| | `google/gemma-3-27b-it` |

We benchmark open and closed-weight widely used VLMs on RAVENEA, leveraging various retrievers against non-RAG baselines, assessing retrieval effectiveness across models of different sizes. The open-sourced models include LLaVA-OneVision-7B (Li et al., 2025), Pixtral-12B (Agrawal et al., 2024), Phi-4 Multimodal-Instruct (Abouelenin et al., 2025), Gemma3-4B-Instruct and 27B-Instruct (Team et al., 2025), Qwen2.5-VL-Instruct (3B, 7B, 72B) (Bai et al., 2025b), Qwen3-VL-Instruct(3B, 7B, 32B), InternVL3 (2B, 8B, 78B) (Zhu et al., 2025), and Deepseek-VL2 variants (Tiny and Base) (Wu et al., 2024). For the closed models, we adopt GPT-4.1 (Achiam et al., 2023) (accessed on 2025/04/14).[8] For proprietary model, GPT-4.1, we directly call the corresponding API. For open-sourced models, we use vllm (Kwon et al., 2023). During the evaluation, to ensure the stability of the results, we set the temperature parameter to 0.0 and the maximum output length to 2048. All open-weight models are listed in Table 10. To avoid the impact of the length the retrieved content, we use the first 256 words in the top-1 Wikipedia document.

## J MORE RESULTS

As illustrated in Figures 11a and 11b, the models exhibit diverse cultural preferences. Notably, most models achieve relatively stronger performance on Indian cultural contexts in both cIC and

---

[8]Knowledge cutoff: June 1, 2024; `https://platform.openai.com/docs/models/gpt-4.1`

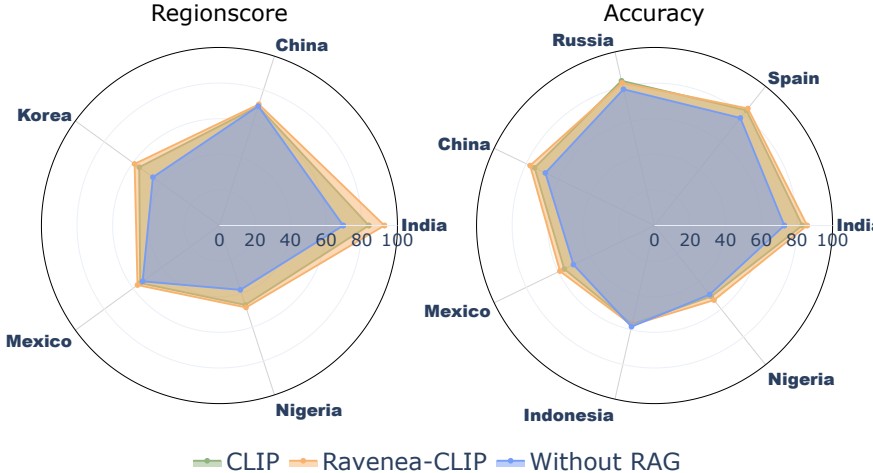

Figure 10: Average improvement across 16 open-source VLMs for different countries with 2 retrievers on both cIC RegionScore and cVQA accuracy.

cVQA tasks. As shown in Figures 10, VLMs demonstrate varying degrees of performance shifts across cultural contexts in both the cIC and cVQA tasks. On cIC, `Ravenea-CLIP` yields noticeable improvements over their original counterparts in Korea, India and Nigeria, indicating the effectiveness of retrieval adaptation. On cVQA, while `Ravenea-CLIP` generally exhibit large performance improvement compared to non-RAG baselines, certain countries, such as Indonesia, experience exhibit diminishing returns.

Table 11: **Ablation at cultural context.** Models in gray are frozen retrievers, noisy characters, and unrelated documents. VLMs augmented with finetuned retriever generally perform better.

| Retriever | Average | DeepSeek-VL2-Tiny | Qwen2.5-VL-3B | Qwen2.5-VL-7B | Qwen2.5-VL-72B | InternVL3-2B | InternVL3-8B | InternVL3-78B | Gemma3-4B | Gemma3-27B | Phi4-Multimodal | Pixtral-12B | LLaVA-OV-7B |
|---|---|---|---|---|---|---|---|---|---|---|---|---|---|
| $cVQA_{Accuracy\uparrow}$ | | | | | | | | | | | | | |
| W/O RAG | 66.0 | 49.8 | 62.7 | 52.6 | **83.3** | 64.1 | 71.3 | 84.2 | 66.5 | 79.4 | 42.6 | 73.2 | **62.7** |
| Unrelated documents | 60.6 | 46.4 | 58.9 | 53.6 | 76.1 | 60.8 | 67.0 | 82.3 | 59.8 | 74.6 | 41.1 | 63.2 | 43.5 |
| Random Characters | 62.3 | 44.0 | 57.9 | 52.6 | 79.4 | 63.2 | 69.9 | 83.7 | 54.1 | 75.1 | 41.1 | 68.9 | 57.4 |
| CLIP-L/14@224px | 69.4 | 50.2 | 63.6 | 66.0 | 82.8 | 73.7 | 78.5 | 84.2 | 69.4 | 76.6 | 67.0 | **76.6** | 44.5 |
| **Ravenea-CLIP (ours)** | **71.2** | 50.7 | 64.1 | 69.9 | 82.3 | 76.1 | 81.3 | 85.2 | 69.9 | 79.9 | 69.4 | 75.1 | 50.2 |
| $cIC_{RegionScore\uparrow}$ | | | | | | | | | | | | | |
| W/O RAG | 54.4 | 26.5 | 53.1 | 69.4 | 71.4 | 38.8 | 49.0 | 65.3 | 73.5 | **75.5** | 28.6 | 53.1 | 49.0 |
| Unrelated documents | 48.8 | 18.4 | 46.9 | 57.1 | 61.2 | 28.6 | 51.0 | 63.3 | 59.2 | 71.4 | 24.5 | 51.0 | 53.1 |
| Random Characters | 51.2 | 24.5 | 44.9 | 61.2 | 67.3 | 36.7 | 44.9 | 67.3 | 61.2 | 73.5 | 28.6 | 59.2 | 44.9 |
| CLIP-L/14@224px | 64.1 | 38.8 | 59.2 | 71.4 | 69.4 | 65.3 | 65.3 | 71.4 | 73.5 | 73.5 | 61.2 | 69.4 | 51.0 |
| **Ravenea-CLIP (ours)** | **67.7** | 46.9 | 61.2 | 77.6 | 75.5 | 69.4 | 67.3 | 73.5 | 75.5 | 75.5 | 63.3 | 71.4 | 55.1 |

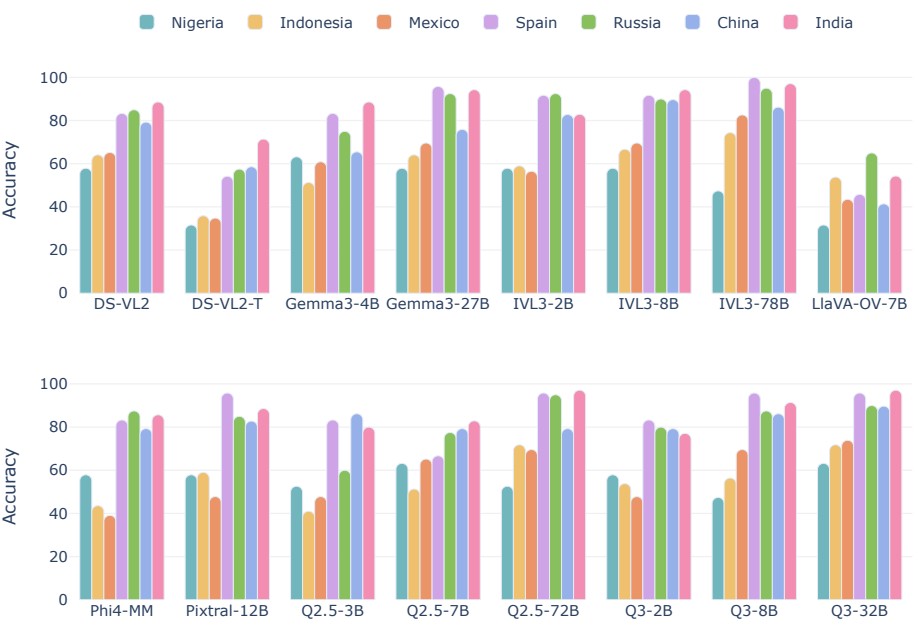

(a) Performance of the 16 VLMs equipped with `Ravenea-CLIP` in cVQA task.

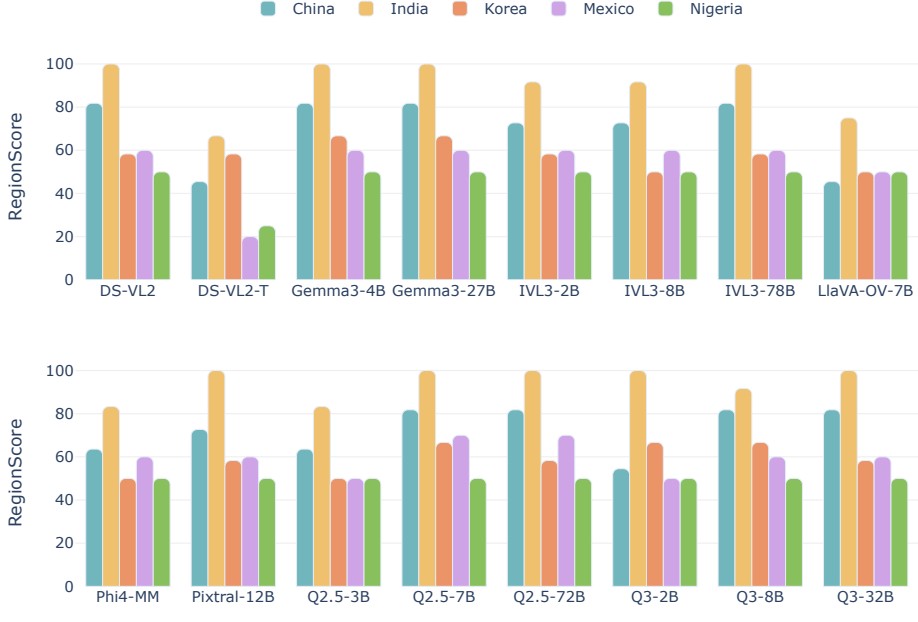

(b) Performance of the 16 VLMs equipped with `Ravenea-CLIP` in cIC task.

Figure 11: Performance of the 16 VLMs equipped with `Ravenea-CLIP` across tasks and countries.

Table 12: Four metrics comparison of RAG and Non-RAG methods for CCUB task.

| Method | Average | DeepSeek-VL2-Tiny | DeepSeek-VL2 | Qwen2.5-VL-3B | Qwen2.5-VL-7B | Qwen2.5-VL-72B | InternVL3-2B | InternVL3-8B | InternVL3-38B | Gemma3-4B | Gemma3-27B | Phi4-Multimodal | Pixtral-12B | LLaVA-OV-7B |
|---|---|---|---|---|---|---|---|---|---|---|---|---|---|---|
| $CCUB_{Rouge-L}$ | | | | | | | | | | | | | | |
| **W/O RAG** | 18.0 | 19.4 | 18.1 | 24.3 | 18.3 | 17.1 | 17.4 | 18.6 | 17.0 | 15.3 | 14.4 | 15.2 | 15.1 | 23.6 |
| **Frozen Models** | | | | | | | | | | | | | | |
| SigLIP2-SO/14@384px | 14.9 | 11.6 | 16.9 | 18.7 | 16.1 | 14.8 | 14.0 | 15.3 | 16.2 | 13.0 | 12.6 | 12.4 | 15.0 | 17.3 |
| CLIP-L/14@224px | 15.3 | 13.6 | 18.1 | 19.8 | 17.0 | 14.8 | 13.1 | 15.1 | 15.5 | 12.9 | 13.8 | 12.9 | 15.0 | 17.3 |
| **Finetuned Models** | | | | | | | | | | | | | | |
| VisualBERT | 14.6 | 11.1 | 16.1 | 17.4 | 15.7 | 14.7 | 12.2 | 15.3 | 15.6 | 12.9 | 13.7 | 11.8 | 15.8 | 18.0 |
| VL-T5 | 14.4 | 10.6 | 15.9 | 18.0 | 16.8 | 14.3 | 13.0 | 15.1 | 15.6 | 12.6 | 13.1 | 10.1 | 14.8 | 17.2 |
| LLaVA-OV-7B | 14.0 | 10.0 | 15.5 | 17.9 | 16.1 | 14.4 | 12.4 | 14.3 | 15.8 | 12.5 | 12.7 | 10.2 | 14.1 | 16.5 |
| Ravenea-SigLIP | 14.8 | 11.7 | 16.4 | 17.3 | 15.5 | 15.5 | 13.3 | 16.1 | 15.6 | 13.2 | 13.0 | 13.0 | 15.1 | 16.9 |
| Ravenea-CLIP | 15.3 | 12.1 | 16.7 | 19.5 | 18.1 | 15.6 | 12.8 | 15.6 | 16.1 | 12.5 | 13.3 | 13.6 | 14.7 | 18.3 |
| $CCUB_{CIDER}$ | | | | | | | | | | | | | | |
| **W/O RAG** | 22.2 | 37.4 | 26.1 | 60.7 | 19.1 | 11.0 | 5.7 | 20.2 | 10.8 | 7.5 | 6.5 | 6.3 | 18.0 | 59.2 |
| **Frozen Models** | | | | | | | | | | | | | | |
| SigLIP2-SO/14@384px | 8.5 | 18.3 | 12.5 | 17.5 | 9.8 | 3.5 | 2.0 | 5.6 | 3.0 | 1.4 | 2.0 | 0.6 | 13.1 | 20.5 |
| CLIP-L/14@224px | 9.0 | 21.5 | 17.0 | 18.9 | 11.5 | 4.3 | 0.6 | 3.2 | 3.3 | 3.4 | 2.7 | 1.6 | 10.6 | 18.0 |
| **Finetuned Models** | | | | | | | | | | | | | | |
| VisualBERT | 9.5 | 19.3 | 16.0 | 19.9 | 9.2 | 4.1 | 0.5 | 6.0 | 1.8 | 3.1 | 6.1 | 1.3 | 18.1 | 18.6 |
| VL-T5 | 7.2 | 16.1 | 12.6 | 10.1 | 8.9 | 3.2 | 0.2 | 3.5 | 0.7 | 3.5 | 3.4 | 0.9 | 13.3 | 17.5 |
| LLaVA-OV-7B | 7.1 | 17.9 | 10.5 | 21.6 | 6.1 | 0.7 | 1.2 | 4.2 | 0.1 | 2.4 | 0.4 | 1.2 | 14.4 | 11.3 |
| Ravenea-SigLIP | 8.0 | 16.8 | 12.9 | 12.5 | 10.0 | 5.4 | 0.4 | 5.3 | 3.6 | 1.8 | 1.8 | 0.7 | 13.3 | 19.0 |
| Ravenea-CLIP | 8.5 | 16.8 | 11.8 | 17.3 | 10.7 | 5.0 | 0.1 | 4.5 | 3.5 | 1.3 | 1.9 | 1.1 | 12.6 | 23.3 |
| $CCUB_{BERTScore}$ | | | | | | | | | | | | | | |
| **W/O RAG** | 53.6 | 53.3 | 55.3 | 56.2 | 54.5 | 53.2 | 51.7 | 54.2 | 53.6 | 50.1 | 51.4 | 51.9 | 53.5 | 57.5 |
| **Frozen Models** | | | | | | | | | | | | | | |
| SigLIP2-SO/14@384px | 50.9 | 46.7 | 52.3 | 54.2 | 53.1 | 50.4 | 48.9 | 51.8 | 51.9 | 48.1 | 50.3 | 49.3 | 51.2 | 53.7 |
| CLIP-L/14@224px | 51.9 | 49.7 | 53.1 | 54.5 | 53.7 | 51.6 | 49.4 | 52.8 | 52.5 | 49.2 | 50.9 | 49.8 | 52.7 | 55.0 |
| **Finetuned Models** | | | | | | | | | | | | | | |
| VisualBERT | 50.6 | 46.6 | 52.4 | 53.9 | 52.8 | 50.2 | 48.4 | 51.9 | 51.3 | 47.8 | 50.1 | 48.1 | 51.4 | 53.5 |
| VL-T5 | 50.2 | 46.2 | 51.9 | 54.4 | 52.8 | 50.0 | 47.7 | 51.0 | 51.3 | 47.7 | 49.7 | 46.8 | 50.7 | 52.8 |
| LLaVA-OV-7B | 50.2 | 46.1 | 51.4 | 54.5 | 52.4 | 49.5 | 47.7 | 51.2 | 51.5 | 47.3 | 49.6 | 47.4 | 50.4 | 53.3 |
| Ravenea-SigLIP | 51.2 | 46.7 | 53.0 | 52.7 | 52.8 | 51.7 | 49.1 | 53.0 | 52.6 | 48.9 | 50.5 | 49.3 | 51.6 | 54.1 |
| Ravenea-CLIP | 52.5 | 49.2 | 54.2 | 54.9 | 54.2 | 52.9 | 49.3 | 53.2 | 53.0 | 49.0 | 51.4 | 51.3 | 53.4 | 56.4 |
| $CCUB_{CLIPScore}$ | | | | | | | | | | | | | | |
| **W/O RAG** | 33.4 | 33.3 | 34.1 | 32.5 | 33.8 | 33.4 | 34.1 | 34.3 | 34.5 | 33.9 | 33.8 | 32.4 | 32.6 | 32.2 |
| **Frozen Models** | | | | | | | | | | | | | | |
| SigLIP2-SO/14@384px | 32.7 | 29.6 | 32.9 | 33.4 | 32.7 | 33.0 | 33.8 | 33.0 | 33.7 | 32.1 | 33.7 | 31.8 | 31.9 | 33.3 |
| CLIP-L/14@224px | 32.7 | 31.4 | 33.4 | 33.7 | 33.2 | 33.7 | 35.1 | 34.1 | 34.1 | 33.2 | 34.1 | 32.7 | 32.5 | 33.6 |
| **Finetuned Models** | | | | | | | | | | | | | | |
| VisualBERT | 32.7 | 29.4 | 32.8 | 33.7 | 32.7 | 33.1 | 33.8 | 33.1 | 33.8 | 32.0 | 33.8 | 31.6 | 31.8 | 33.2 |
| VL-T5 | 32.8 | 29.7 | 33.0 | 33.4 | 32.8 | 33.0 | 33.7 | 33.3 | 33.7 | 32.0 | 33.9 | 31.7 | 31.8 | 33.2 |
| LLaVA-OV-7B | 32.7 | 29.6 | 32.8 | 33.6 | 32.7 | 32.9 | 34.0 | 33.1 | 33.6 | 32.0 | 33.8 | 31.7 | 31.9 | 33.0 |
| Ravenea-SigLIP | 32.7 | 29.8 | 33.0 | 32.1 | 33.1 | 33.5 | 33.8 | 33.1 | 33.8 | 32.7 | 33.7 | 31.7 | 32.0 | 33.1 |
| Ravenea-CLIP | 33.5 | 31.6 | 33.4 | 34.0 | 33.6 | 33.7 | 34.7 | 34.2 | 34.2 | 33.0 | 33.9 | 32.8 | 32.8 | 33.3 |

## K  PROMPTS

The prompts that we used to collect the documents and two downstream tasks were designed to ensure consistency across different models. One prompt example for cultural caption generation is shown in Table 13. The prompts for both with and without retrieval augmentation for VLMs are shown in Table 15 and 14.

Table 13: **Prompt example for generating captions used to retrieve the Wiki documents.**

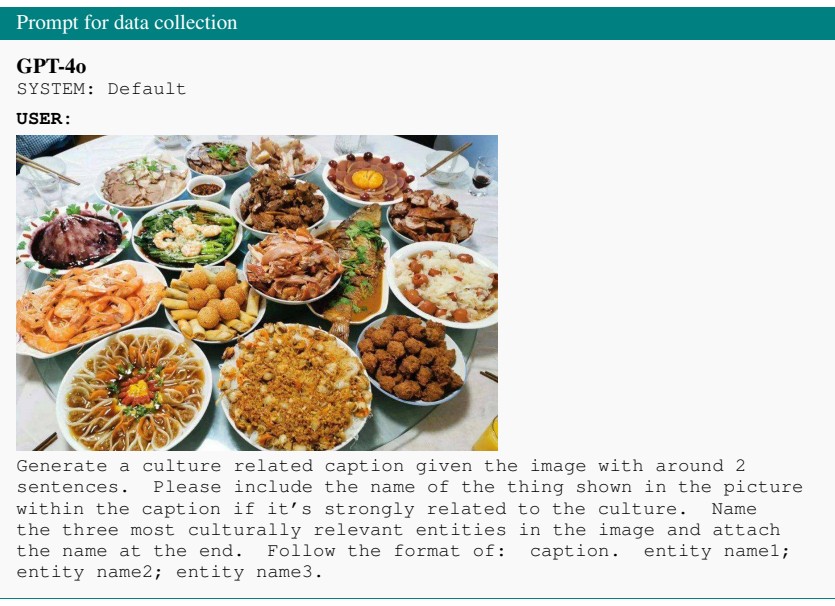

Table 14: **Prompt examples without RAG.** Multimodal prompt samples with interleaved image are shown for both CVQA and CCUB tasks.

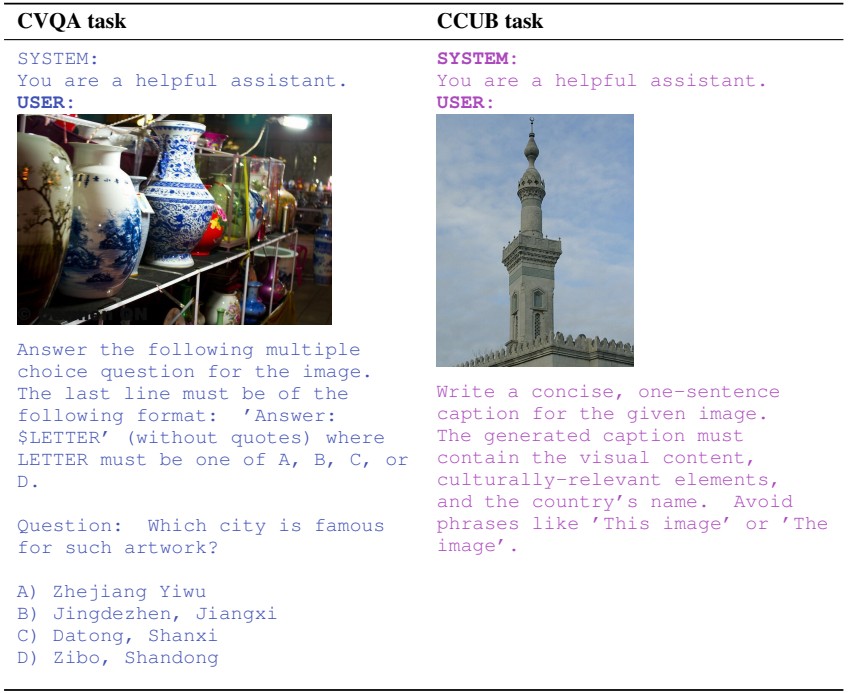

Table 15: **Prompt examples `with` RAG.** Multimodal prompt samples with interleaved image are shown for both CVQA and CCUB tasks.

| CVQA task | CCUB task |
|---|---|
| SYSTEM:
You are a helpful assistant.
USER:
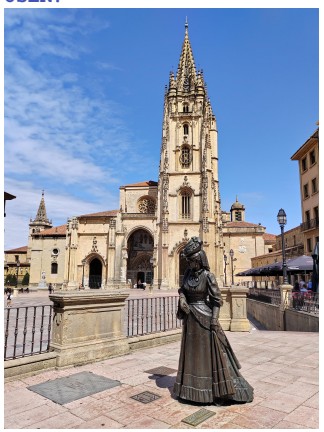
Answer the following multiple choice question for the image. The last line must be of the following format: 'Answer: $LETTER' (without quotes) where LETTER must be one of A, B, C, or D.
Use the shared culture between the image and the following document to answer the question.

Document:
Buildings and structures Buildings about 800 – Borobudur temple in Java completed. 802 Haeinsa of Korea, is constructed. Palace of Charlemagne in Aachen, Carolingian Empire completed (begun about 790). The Palatine Chapel still stands. At Oviedo in the Kingdom of Asturias Cámara Santa constructed. First reconstruction of Oviedo Cathedral begun by Tioda. 815 – Second Temple of Somnath built in the Pratihara Empire, India. 816 – Reims Cathedral begun. 810s – Chapel of San Zeno in Santa Prassede, Rome decorated. 818 – Old Cologne Cathedral built....

Question: What is the name of the square where the cathedral and the statue of the image are located?

A) Riego Square
B) Trascorrales Square
C) The square of Alfonso II the Chaste
D) The Fontan square | SYSTEM:
You are a helpful assistant.
USER:
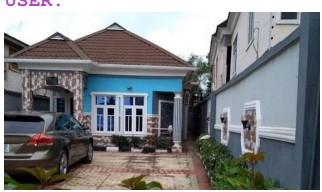
Write a concise, one-sentence caption for the given image. The generated caption must contain the visual content, culturally-relevant elements, and the country's name. Avoid phrases like 'This image' or 'The image'. Please consider the following DOCUMENT as supplementary material for the image. Mention the country's name of the culture.

Document:
Architecture of Nigeria was historically influenced by environmental conditions as well as social and cultural factors. The coming of missionaries and political changes brought about by colonialism precipitated a change in architectural style and utility of buildings. A Gothic revival style was adopted for early churches built in the colony of Lagos. A one or two storey timber house building made with pre-fabricated material components and designed with the influence of classic antiquity styles served as mission house for the missionaries... |

## L HUMAN ANNOTATION & EVALUATION INTERFACE

The annotation interfaces are shown in Figure 13 and 14. The interface for human evaluation in cIC task is shown in Figure 12.

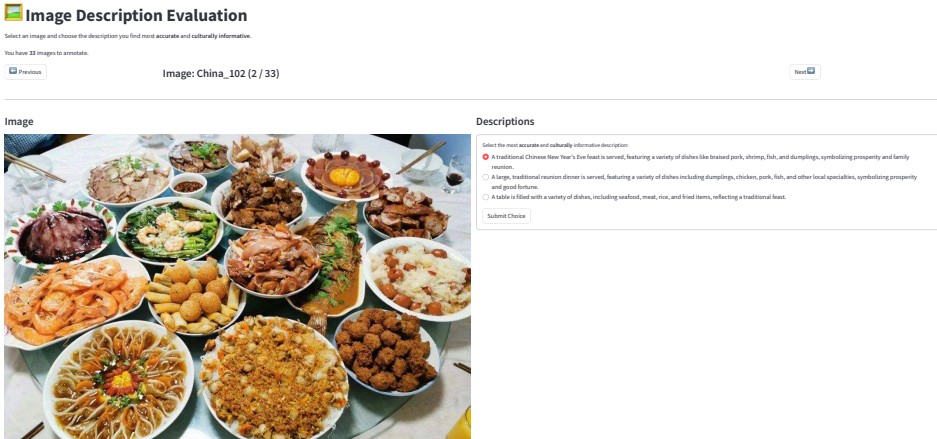

Figure 12: Human evaluation interface for cIC task.

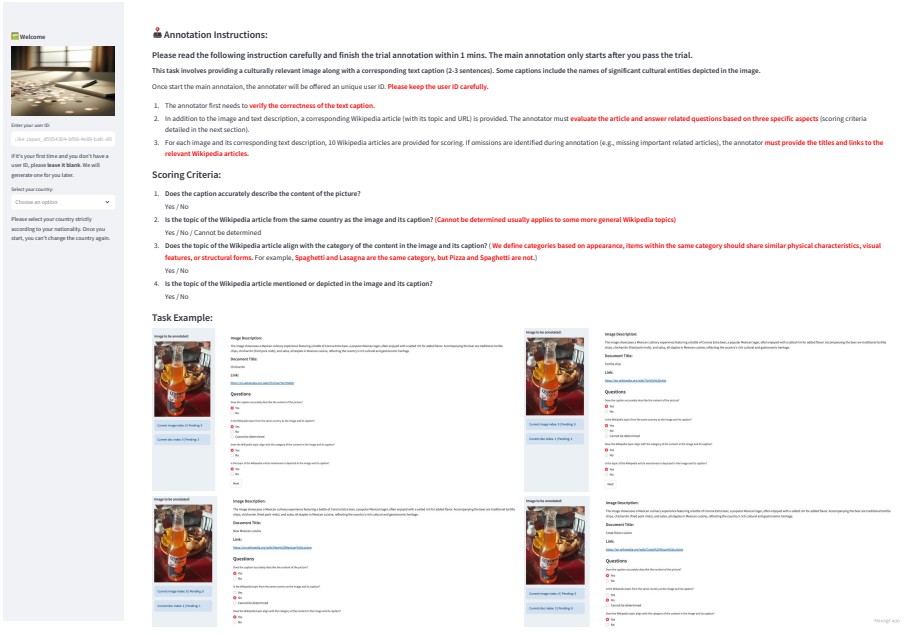

Figure 13: Human annotation instructions.

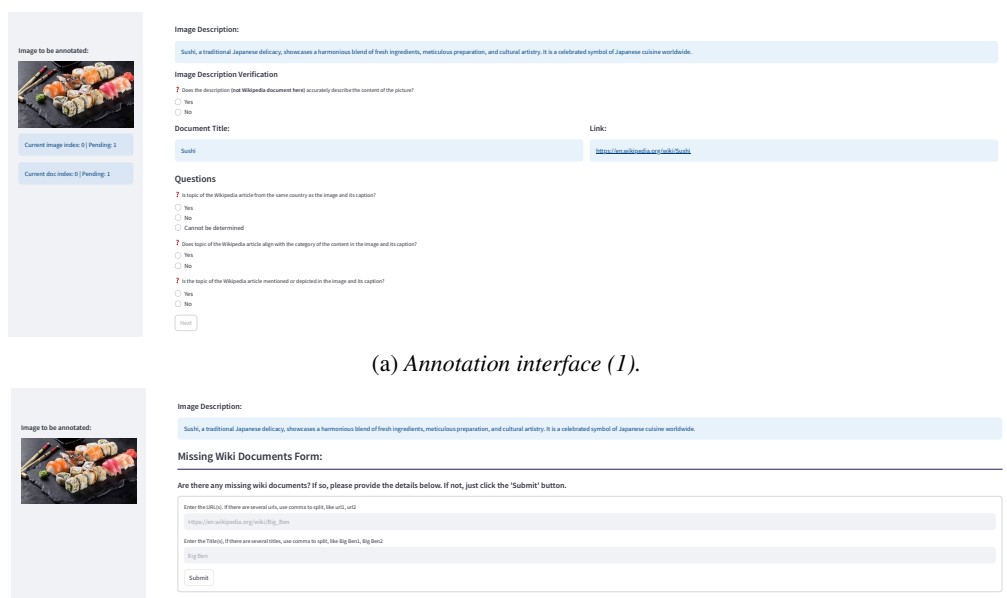

(a) *Annotation interface (1).*

(b) *Annotation interface (2).*

Figure 14: Human annotation interfaces.

