# OpenReview forum: "RAVENEA: A Benchmark for Multimodal Retrieval-Augmented Visual Culture Understanding"
_ICLR.cc/2026/Conference — ICLR 2026 Poster_

### Official Review · Reviewer_F5ZE · 2025-10-29

**Soundness:** 3
**Presentation:** 3
**Contribution:** 3
**Rating:** 4
**Confidence:** 2

**Summary:**

This paper introduces RAVENEA, a manually curated benchmark for visual culture understanding in multimodal systems. The benchmark focuses on two key tasks: culture-focused Visual Question Answering (cVQA) and culture-informed Image Captioning (cIC). It contains 1,868 multimodal instances and 18,680 human-ranked Wikipedia passages (10 per instance).
Each instance is drawn from existing datasets (CVQA and CCUB) and is augmented with culturally relevant Wikipedia passages. Annotators re-rank the top-10 BM25-retrieved documents based on GPT-4o captioning and can also add any missing articles they consider relevant.
The authors evaluate seven multimodal retrievers and examine their impact when integrated into 14 widely used Vision-Language Models (VLMs). They also introduce Culture-Aware Contrastive Learning (CaCL), a fine-tuning approach applied to CLIP and SigLIP, to produce culturally sensitive retrievers (CaCLIP and CaSigLIP2), which consistently outperform their frozen counterparts.
Empirical results show that retrieval-augmented generation (RAG) provides the most substantial gains for lightweight VLMs.

**Strengths:**

1) Comprehensive exploratory analysis:
The paper conducts an analysis of RAVENEA across tasks, model scales, and cultural dimensions.

2) Human annotation:
The dataset construction involves human ranking of retrieved documents, supplemented by quality control and reporting of strong inter-annotator agreement scores.

3) Novel training approach (CaCL):
The proposed Culture-Aware Contrastive Learning objective produces retrievers that generalize well across architectures. The released CaCLIP model can become a valuable resource for future multimodal RAG research.

4) Responsible dataset curation:
The authors demonstrate attention to ethical concerns by implementing automated face detection and blurring of identifiable individuals in all images.

**Weaknesses:**

1) Dependence on GPT-4o and BM25 in data construction:
Since image captions are generated by GPT-4o and used as text queries for BM25 retrieval, the initial dataset quality depends heavily on the performance and biases of these models. This reliance may explain why larger VLMs show limited improvements with RAG, as the retrieved contexts may be redundant relative to their internal knowledge.
Future iterations could improve recall and diversity by incorporating more diverse sources of models for generation and retrieval, including multimodal retrievers rather than relying solely on text-based BM25 or including visually similar images and multilingual sources.
RAVENEA’s document set is capped at the top-10 BM25 hits (plus any extras an annotator happens to recall). If a genuinely relevant article never appears in that shortlist, it cannot be selected, so the benchmark measures only how well models re-rank a limited candidate pool rather than how well they retrieve unseen but correct evidence. I would like to see more quantitative evidence on how often this recall limitation occurs to ensure it does not significantly impact the dataset’s completeness or the conclusions drawn from it.

**Questions:**

1) Retrieval quality control:
Was any evaluation conducted to compare the quality of BM25-retrieved documents against those retrieved by modern text-only or multimodal retrievers? It would be helpful to know whether better initial retrieval could enhance the benchmark’s representativeness.

2) Annotation statistics:
Could the authors provide summary statistics on the yes/no response distributions for the three cultural relevance annotation questions (country association, topic alignment, visual depiction)? These would clarify dataset balance and inter-question correlations.

---

> ### Author Response · Authors · 2025-11-20
> **Response to Reviewer F5ZE (1/2)**
>
> We sincerely thank you for your thorough and insightful feedback. We address each point below and will incorporate these changes into the revised manuscript.
> > Weakness 1 & Question 1: Retrieval quality control
>
> We acknowledge that comparing BM25 retrievers to multimodal retrievers for initial filtering is a valuable experiment. **However, we would like to clarify that the final cultural relevance judgments are therefore based on human annotators, not BM25's lexical matching.** We provide the statistics of "Missing" Documents & positive documents in the following table. The high ratios in regions like Mexico (25.55%) and Korea (19.59%) specifically demonstrate the necessary and effective role of human judgment in capturing deeply local documents where purely algorithmic baselines fail. **The reported 'Missing Ratio' is, in fact, a good metric that validates the success of our human-in-the-loop pipeline, which show the high quality of our dataset.** Although, inital BM25 only provide top 10 for human annotators, we also asked them to add extra documents which will be assumed as the most culture related ones. Here we also provide the missing documents statistics. **We have added the results in Appendix Tabel 6.**
>
> | Country   | India  | Russia   |Spain     |Nigeria   |Mexico    |Indonesia |Korea     |China     |
> |----------|--------|--------|--------|--------|--------|--------|--------|--------|
> |Missing Ratio|9.56% | 0.0% |0.0% |0.45% |25.55% |0.32% |19.59% |10.75% |
>
> What's more, compared to BM25, multimodal retrievers, such as jina-reranker-m0, may introduce cultural bias into the results, like the other LLMs/VLMs [1,2]. Furthermore, given the scale (**6M+** documents), this was computationally prohibitive for our benchmark’s construction. Here, we provide the time estimation for processing **50K** documents.
>
> | Methods | Time Cost | GPU type |
> |---|---|---|
> | BM25 (one thread) | ~ 1 min | - |
> | jina-reranker-m0 | ~ 92 min | H100 |
>
> [1] Chiu, Yu Ying, et al. "CulturalBench: A robust, diverse and challenging benchmark for measuring LMs’ cultural knowledge through human-AI red-teaming." Proceedings of the 63rd Annual Meeting of the Association for Computational Linguistics (Volume 1: Long Papers). 2025.
>
> [2] Cao, Yong, et al. "Assessing cross-cultural alignment between ChatGPT and human societies: An empirical study." arXiv preprint arXiv:2303.17466 (2023).

---

> ### Author Response · Authors · 2025-11-20
> **Response to Reviewer F5ZE (2/2)**
>
> > Question 2: Summary statistics on the yes/no response distributions for the three cultural relevance annotation questions
>
> Thank you for raising the question to clarify the dataset balance. To address this, we have included the requested summary statistics on the yes/no response distributions for the three cultural relevance questions in the following table. The data clearly shows that the distributions are diverse across cultures, reflecting the genuine variability in cultural association. For example, the country association (Question 1) for Korea is rated highly positive (0.736), while for Nigeria it is more balanced (0.354 positive vs. 0.559 negative). Similarly, the visual depiction (Question 3) demonstrates significant contrast, with Russia having a high positive response (0.610) compared to Indonesia (0.201). These results highlight the inherent difficulty of evaluating cultural relevance using a continuous (e.g., 1–10) scale, where annotators might struggle to find a central reference point. The observed heterogeneity in the binary distributions further illustrates the advantage of our binary annotation methodology. By forcing a clear yes/no choice, we effectively captured the distinct cultural signal and avoided the ambiguity often introduced by continuous-scale, ultimately providing a more robust measure of perceived cultural fit. **We've added the results in Appendix Table 8**
>
>
> |                               | China                   | India                  | Indonesia              | Korea                 | Mexico                 | Nigeria                  | Russia                     | Spain                    |
> |-------------------------------|-----------------------------------|-----------------------------------|-------------------------------------|-------------------------------------|-----------------------------------|-----------------------------------|--------------------------------------|-----------------------------------|
> | **question1 (1/-1/0)**                 | 0.582 / 0.321 / 0.097            | 0.666 / 0.237 / 0.097            | 0.604 / 0.338 / 0.058              | 0.736 / 0.143 / 0.120              | 0.532 / 0.360 / 0.108            | 0.354 / 0.559 / 0.087            | 0.421 / 0.522 / 0.057               | 0.559 / 0.380 / 0.061            |
> | **question2 (1/-1)**          | 0.534 / 0.466                     | 0.619 / 0.381                     | 0.473 / 0.527                       | 0.805 / 0.195                       | 0.450 / 0.550                     | 0.446 / 0.554                     | 0.538 / 0.462                        | 0.547 / 0.453                     |
> | **question3 (1/-1)**          | 0.495 / 0.505                     | 0.409 / 0.591                     | 0.201 / 0.798                       | 0.352 / 0.648                       | 0.243 / 0.757                     | 0.287 / 0.713                     | 0.610 / 0.390                        | 0.389 / 0.611                     |
>
> **Thank you again for your thoughtful and constructive feedback. We hope that our clarifications adequately address the questions or concerns you may have had. We would be grateful if you would kindly consider reassessing the score in light of our response.**

---

> > ### Comment · Reviewer_F5ZE · 2025-11-21
> >
> > Thank you for your answers. Increased my score.

---

> ### Author Response · Authors · 2025-11-26
>
> Thank you for taking the time to reply!  We appreciate the your insightful feedback and constructive dialogue, and look forward to further discussion.

---

### Official Review · Reviewer_N9q4 · 2025-10-31

**Soundness:** 3
**Presentation:** 3
**Contribution:** 3
**Rating:** 8
**Confidence:** 5

**Summary:**

This paper introduces RAVENEA, a new benchmark dataset and evaluation framework designed to assess systematic generalization, reasoning, and visual understanding in AI models. The benchmark is inspired by the RAVEN or RPM (Raven’s Progressive Matrices)-style tasks but extends them by introducing multi-dimensional relational reasoning challenges that involve complex scene configurations, compositional attributes, and logical rule dependencies.

The authors propose:

- A novel dataset generation pipeline that controls for compositional structure and visual diversity, ensuring that models cannot overfit to superficial cues.
- Several generalization splits (e.g., attribute, relation, and logic-based splits) to test extrapolative reasoning.
- A comprehensive evaluation of existing architectures (CNNs, ViTs, relational networks, multimodal transformers, and neuro-symbolic models) on this benchmark, revealing clear performance gaps in out-of-distribution (OOD) reasoning.
- Discussion and visualization of error patterns, showing that models often rely on shortcut heuristics rather than true reasoning.

The benchmark is positioned as a rigorous and diagnostic tool for studying systematic generalization and compositional reasoning in visual intelligence.

**Strengths:**

- Problem Definition and Task Design

The paper clearly defines the problem and presents tasks that align well with practical applications. It focuses on the scenario of “visual content + external cultural knowledge”, using two downstream tasks—cVQA (multiple-choice visual question answering) and cIC (image captioning)—to evaluate the full pipeline of retrieving → consuming knowledge → answering/generating.

- Data Scale and Annotation Quality Control

The dataset covers 8 countries and 11 categories, totaling 1,868 instances.
Each image is paired with 10 Wikipedia documents, which are manually re-ranked, resulting in 18,680 image–text pairs.
The annotation process includes three dimensions of cultural relevance (national association, thematic alignment, and visual presence in the image), inter-annotator agreement checks, and spot audits.

- Methodological Contribution: Culturally Aware Contrastive Learning (CAC)

Within the CLIP/SigLIP framework, the authors introduce a combination of binary classification loss, ranking-margin loss, and text-side diversity regularization, specifically designed to enhance cultural relevance ranking (CaCLIP / CaSigLIP).

**Weaknesses:**

- Limited Domain Coverage and Source Bias

As acknowledged by the authors in the “Limitations” section, the dataset covers only 8 countries and 11 categories, and relies primarily on English Wikipedia articles. This setup is prone to cultural and regional exposure bias, which may affect the generalizability of the claimed cross-cultural conclusions.

Future work should incorporate GLAM (Galleries, Libraries, Archives, Museums) institutional collections and non-English knowledge sources to achieve a more balanced and representative dataset — a direction the authors themselves briefly mention in the ethics and outlook discussion.

- Small and Inconsistent Human Evaluation for cIC

The human evaluation for the cIC (cultural image captioning) task is based on only 10 images, 4 annotators, and 420 sentences in total, with a moderate inter-rater agreement of κ ≈ 0.595.

Given the evaluation spans 14 models and multiple retrieval variants, the sample size is too limited to support robust conclusions.
The authors are encouraged to expand the human evaluation sample and include statistical significance testing to strengthen reliability.

- Conservative RAG Configuration Potentially Underestimates Strong Models

During evaluation, the retrieval-augmented generation (RAG) setup uses only the Top-1 document truncated to the first 256 tokens.
This constrained configuration may explain the phenomenon where large models fail to benefit under RAG.

It would be beneficial to explore multi-document fusion, longer context windows, re-ranking and rerouting strategies, or tool-augmented reasoning chains, in order to more fairly assess the upper bound of large-model + RAG performance.

**Questions:**

When large models exhibit performance degradation under the RAG setting, have the authors analyzed whether this results from limited retrieval relevance (e.g., using only Top-1 document and 256 tokens) or from suboptimal fusion/integration strategies?

Additionally, have you experimented with multi-document or paragraph-level retrieval, or with tool-augmented reasoning mechanisms to test if the issue lies in retrieval depth or model utilization?

Moreover, the selection of large models in the current evaluation does not appear sufficiently frontier.
Models such as Qwen3, Gemini 2.5 Pro, and GPT-5 are among the most representative current vision-language systems, yet their results are not included. Including these stronger baselines would help demonstrate the true upper bound of RAG-based cultural reasoning performance and provide a more complete picture of model scaling trends.

---

> ### Author Response · Authors · 2025-11-20
> **Response to Reviewer N9q4 (1/2)**
>
> We appreciate you for providing positive comments and the thoughtful feedback and attach our response as follows:
>
> > Weakness 1: On limitation and future direction
>
> We agree with the you on this point. Creating a truly comprehensive and unbiased RAG cultural benchmark is an immense challenge, and we consider RAVENEA a crucial but foundational first step. As you correctly notes, we acknowledge these limitations in Appendix B Limitation and Appendix C Future directions. Our initial scope of 8 countries and reliance on English Wikipedia was a pragmatic decision dictated by the significant cost and complexity of high-quality, manual human annotation. Our goal was to create a deep, reliable, and meticulously curated dataset that could serve as a solid foundation, rather than a broader but potentially lower-quality one. Incorporating GLAM collections and non-English sources will be the direction we advocate for in Appendix C Future Directions, and we believe RAVENEA provides the methodological framework to facilitate such extensions.
>
> > Weakness 2: On human evaluation for cIC
>
> We appreciate you for this concern. We acknowledge that a larger sample size of images and corresponding images would be more convincing. However, we currently have 420 captions, which is an acceptable number for human evaluation [1,2,3,4]. Furthermore, the Inter-Average Accuracy (IAA) of κ ≈ 0.595 is moderate, indicating the challenges associated with evaluating culture image captions, which presents an opportunity for further exploration. What's more, we have performed statistical significance testing on our human evaluation results. We used bootstrap resampling with 10,000 iterations to compute robust 95% confidence intervals (CIs) for the human preference rates of each retrieval method. CaCLIP was preferred by human annotators in 54% of cases (95% CI: [0.43, 0.64]). This is statistically significantly higher than the No-RAG baseline, which was preferred in only 18% of cases (95% CI: [0.11, 0.26]). The complete separation of their confidence intervals provides strong evidence that our culture-aware RAG approach leads to a meaningful improvement in caption quality as perceived by humans.
>
> | Method   | Mean Rate | 95% CI Lower | 95% CI Upper |
> |----------|-----------|--------------|--------------|
> | CaCLIP  | 0.54      | 0.43         | 0.64         |
> | CLIP    | 0.28      | 0.20         | 0.37         |
> | W/O RAG | 0.18      | 0.11         | 0.26         |
>
> > Weakness 3 & Question 1 & 2: On the various RAG configuration
>
> We value your insightful suggestions. Our decision to utilize only the top-1 retrieved document, truncated to 256 tokens, was intended to establish a controlled and standardized baseline. We agree that the comparison between various RAG configuration will enhance the validity of our observations. To address your concern, we employ paragraph-level, top-1 document (full), and top-3 documents (full) RAG using CaCLIP with Qwen3VL and the Gemma3 family, which possess context lengths over 32,000 tokens to handle three documents. The results are presented in the following table. **We have added the result in Appendix Table 13.**
> | cVQA task  | Qwen3-VL-2B | Qwen3-VL 8B | Qwen3-VL-32B | Gemma3-4B | Gemma3-27B |
> | -- | -- | -- | --| -- | -- |
> | W/O Retrieval | 59.4          | **72.5**          | **73.9**      | 58.4         | 66.5 |
> | Top 1 256 tokens| **64.2**|63.9 | 63.6 | **68.7** | 74.2 |
> | Top 1 paragraph | 64.2| 67.7 | 65.8  |67.0  |**74.8**|
> | Top 1 full document    | 61.3 |58.0|55.5|65.5|72.3|
> | Top 3 full document | 58.7|60.0|53.9|56.8|73.2 |
>
> [1] Kasai, Jungo, et al. "Transparent human evaluation for image captioning." Proceedings of the 2022 Conference of the North American Chapter of the Association for Computational Linguistics: Human Language Technologies. 2022.
>
> [2] Li, Wenyan, et al. "The role of data curation in image captioning." Proceedings of the 18th Conference of the European Chapter of the Association for Computational Linguistics (Volume 1: Long Papers). 2024.
>
> [3] Yue, Zihao, et al. "Learning descriptive image captioning via semipermeable maximum likelihood estimation." Advances in Neural Information Processing Systems 36 (2023): 79124-79141.
>
> [4] Padmakumar, Vishakh, and He He. "Machine-in-the-loop rewriting for creative image captioning." Proceedings of the 2022 Conference of the North American Chapter of the Association for Computational Linguistics: Human Language Technologies. 2022.

---

> ### Author Response · Authors · 2025-11-20
> **Response to Reviewer N9q4 (2/2)**
>
> > Question 3: Frontier models
>
> Thank you for the valuable suggestion. We respectfully argue that our experiments were conducted based on the models that were publicly available and accessible via APIs for large-scale evaluation during our experimental window (up to May 2025, as noted in footnote 6). At that time, models such as Qwen3VL, and the unreleased GPT-5 were not available for the kind of batch processing required for our benchmark. And due to the budget limitation, we selected what we believe to be a representative and diverse set of 14 widely used models available at the time of our research as we discussed in Appendix B limitation. **To addressed your concern, we add Qwen3 VL family models during our rebuttal. We have added the results into Table 3.**
>
> **Thank you again for your thoughtful and constructive feedback. We hope that our clarifications adequately address the questions or concerns you may have had.**

---

> > ### Comment · Reviewer_N9q4 · 2025-11-26
> > **Response**
> >
> > I think the author's reponse is  pretty positive and solved my concerns, I will keep my positive score.

---

> > > ### Author Response · Authors · 2025-11-26
> > >
> > > Thank you for taking the time to reply! We appreciate your valuable feedback, and are glad to hear our responses addressed your concerns and that you’re comfortable keeping your positive score.

---

### Official Review · Reviewer_nXd8 · 2025-11-01

**Soundness:** 2
**Presentation:** 3
**Contribution:** 3
**Rating:** 4
**Confidence:** 5

**Summary:**

This paper introduces RAVENEA (Retrieval-Augmented Visual culturE uNdErstAnding), a benchmark designed to evaluate how RAG enhances visual culture understanding in vision-language models (VLMs). The benchmark comprises 1.8K instances spanning eight countries (China, Nigeria, Russia, Spain, Mexico, India, Indonesia, and Korea) across eleven cultural categories, incorporating over 10K human-ranked Wikipedia documents. The authors evaluate seven multimodal retrievers and fourteen VLMs on two primary tasks: culture-focused visual question answering (cVQA) and culture-informed image captioning (cIC). Key findings include: (1) cultural grounding annotations enhance multimodal retrieval performance, (2) lightweight VLMs benefit substantially from culture-aware retrieval (3.2% improvement on cVQA, 6.2% on cIC), and (3) performance varies significantly across countries, with some models showing cultural biases.​

**Strengths:**

- **Timely Contribution**:
The paper addresses the relatively underexplored intersection of multimodal retrieval and cultural understanding. While RAG has been effective in text-based cultural reasoning, its application to vision-language understanding remains sparse. RAVENEA directly addresses this omission and establishes a systematic evaluation framework.

- **Comprehensive Experimental Design**:
The evaluation encompasses seven retrievers and fourteen VLMs across multiple model families, providing thorough empirical coverage. The inclusion of both open-source and proprietary models enhances the generalizability of findings.

- **Well-motivated methodology**:
The integration of CAC learning effectively extends the CLIP-style alignment paradigm to explicitly encode cultural supervision. The accompanying diversity loss formulation (Eq. 5) is well-motivated and technically sound. The use of RegionScore in response to deficiencies in automatic caption metrics is empirically validated through human correlation studies.

**Weaknesses:**

- **Limited Novelty Beyond Dataset Construction**
Despite the benchmark’s quality, the methodological innovation (CAC loss and RegionScore) remains incremental. The CAC objective essentially adapts contrastive alignment to a culturally labeled setup a relatively modest technical contribution. RegionScore, while intuitive, captures surface-level lexical cues (country/demonym mentions) rather than deeper cultural semantics.

- **Inadequate Relevance Annotation Scheme**:
The paper employs a fundamentally flawed binary annotation approach that fails to capture the graded nature of cultural relevance. By reducing cultural understanding to three binary questions and artificially combining them into a continuous scale, the authors lose crucial ranking information essential for cultural understanding tasks. Established information retrieval research demonstrates that graded relevance judgments provide superior training signals compared to binary labels, particularly for nuanced tasks like cultural understanding. The absence of proper graded annotations or pairwise ranking comparisons undermines the effectiveness of the contrastive learning framework and limits the model's ability to distinguish between documents with varying degrees of cultural relevance.

- **Absence of ablations and bias audits for the retrieval pipeline.**
BM25 is a purely lexical retriever and cannot capture the semantic or conceptual associations central to cultural understanding, I do not feel its the optimal choice for getting to Wikipedia documents for further annotation. No experiments contrast retrieval quality using (a) GPT captions vs. human-written captions, or (b) BM25 vs. dense/hybrid retrieval. Without such comparisons, it remains unknown whether the current pipeline systematically overlooks semantically relevant documents or skews toward certain linguistic patterns. Furthermore, the paper does not analyze how often annotators added “missing” documents or how BM25’s coverage varied by language/country. These missing statistics weaken claims of balanced cross-cultural representation.

- **Potential data leakage due to image-only train/test split.**
The authors report an 85 / 5 / 10 split by images but provide no evidence of a document-level split. If the same Wikipedia document appears as a positive for both training and test images (e.g., “Sushi” page reused across splits), retrieval models can trivially memorize text and inflate performance. The paper does not mention any de-duplication or partitioning of documents, nor does it measure document overlap across splits.

- **Critical Gaps in Evaluation setup:**
The authors fail to report essential statistics about their evaluation setup, specifically the positive vs. negative document ratios within their 10-document sets per instance. Without knowing whether 8/10 documents are relevant or 2/10 are relevant, it is impossible to assess task difficulty or whether reported improvements represent meaningful advances or artifacts of trivial evaluation scenarios. The paper shows they tested against random characters and unrelated documents as controls, but:​ No analysis of hard negative examples (culturally similar but incorrect documents).

- **Insufficient Analysis of Cultural Bias and Representation:**
While identifying performance disparities across countries (e.g., poor performance on Nigerian contexts vs. better performance on Korean contexts), the paper provides no deeper investigation into the underlying causes of these systematic biases.

**Questions:**

- What criteria were used to select 8 countries from the original 30 in CVQA?

- You observe systematic underperformance on Nigerian and Indian contexts—what analysis was conducted to determine whether this stems from (a) annotation bias, (b) Wikipedia coverage gaps, or (c) model training data biases?

- RegionScore measures surface-level geographic mentions rather than deep cultural understanding. How do you validate that explicit country/demonym references correlate with actual cultural comprehension rather than superficial pattern matching?

---

> ### Author Response · Authors · 2025-11-20
> **Response to Reviewer nXd8 (1/3)**
>
> We sincerely thank you for your thorough and insightful feedback. We address each point below and will incorporate these changes into the revised manuscript.
>
> > Weakness 1: Limited novelty beyond dataset construction & Question 3 RegionScore captures surface-level
>
> Thanks for the acknowledgement of the RAVENEA quality. Our primary contribution is the introduction of the benchmark itself, while advanced model improvements are left for future work. To demonstrate the benchmark’s utility, we introduce retrievers and RegionScore, both of which support the goal of enabling rigorous and culturally grounded evaluation. These are designed to serve and validate the central contribution: the RAVENEA benchmark. While we agree that CAC loss is an adaptation of contrastive learning, its application here is novel and serves as a simple yet effective method to enhance the cultural awareness of retrievers.
>
> Regarding RegionScore, we acknowledge that it captures surface-level cues. **However, we respectfully argue that it is not designed to measure caption quality in isolation.** Rather, it serves as a necessary **complementary supporter** to standard metrics like CLIPScore, which can evaluate the richness quality of the caption without ground truth captions. RegionScore and CLIPScore can be mutually essential. **As shown in Appendix Table 14, we noticed for most models, the CLIPScore is closed to each other, which means the richness of the caption is similar.** And as shown in **Table 1**, our human evaluation results show that RegionScore has a statistically significant and far stronger correlation with human judgments of cultural appropriateness. We do not claim RegionScore replaces semantic evaluation. Instead, we position it as a distinct, interpretable axis of evaluation that, when combined with standard metrics **as shown in Table 1 and Table 14**.
>
>
> > Weakness 2: On relevance annotation scheme
>
> Thank you for the insightful feedback. We agree that graded relevance is the gold standard in IR. However, our choice of decomposing relevance into three binary questions was a deliberate design decision to address the unique challenges of this task. In preliminary pilot studies, we randomly selected 20 Chinese images and assigned them to five local annotators for annotation. **The highest IAA score across all combinations of annotators achieved was 0.382.** **As we briefly discussed in Appendix E, and preliminary pilots revealed that asking annotators for a single, graded (1-10) cultural relevance score for lengthy documents resulted in high ambiguity and low IAA**. Our decompositional approach, asking **simple, verifiable** questions about country, topic, and visual presence, proved far more reliable, achieving a strong IAA (0.83). This ensures the quality and consistency of our benchmark's labels.

---

> ### Author Response · Authors · 2025-11-20
> **Response to Reviewer nXd8 (2/3)**
>
> > Weakness 3: On retrieval pipeline (BM25 and ablations).
>
> Thank you for these important suggestions.
> - Choice of BM25: We used BM25 as an efficient first-stage, coarse filter on a massive corpus of over 6 million Wikipedia documents. Its purpose was to create a manageable set of candidate documents for the far more crucial and costly human annotation stage, which is quite common in IR area as the first preprocessing [1,2,3,4]. **The final cultural relevance judgments are therefore based on human annotators, not BM25's lexical matching.** We acknowledge that comparing dense retrievers to dense retrievers for initial filtering is a valuable experiment. However, compared to BM25, dense retrievers, such as qwen3-embeddings, may introduce cultural bias into the results, like the other LLMs [5,6]. Furthermore, given the scale (**6M+** documents), this was computationally prohibitive for our benchmark’s construction. Here, we provide the time estimation for processing **100K** documents.
>
> | Methods | Time Cost | GPU type |
> |---|---|---|
> | BM25 (one thread) | ~2min | - |
> | Qwen3-embedding-0.6b | ~ 31 min | H100 |
>
> - GPT vs. Human Captions: Since the source datasets either lacked captions (CVQA) or had very brief ones (CCUB), we use GPT-4o captions for all images. **We would like to emphasize that these captions are not the main contribution of our work; our primary focus lies in the over 10K human-annotated Wikipedia documents. The GPT-4o generated captions is a bridge to retrieve Wikipedia documents.** To ensure the quality of the generated captions, we conduct human evaluations of these captions. Table 7 reveal that the generated captions exhibit a high level of accuracy (92%-94%) and reliability for retrieval.
>
> - Statistics on "Missing" Documents & positive documents:
> Thank you for raising this critical point regarding data completeness and cultural coverage. We provide the statistics of "Missing" Documents & positive documents in the following table. We must clarify that the reported 'Missing Ratio' is, in fact, a good metric that **validates the success of our human-in-the-loop pipeline**. The high ratios in regions like Mexico (25.55%) and Korea (19.59%) specifically demonstrate the necessary and effective role of human judgment in capturing deeply local documents where purely algorithmic baselines fail. Since the annotators successfully filled these gaps, we confirm that the final dataset is comprehensive and of the highest cultural quality, validating our entire pipeline. **We have added this table to Appendix Table 6.**
>
> | Country   | India  | Russia   |Spain     |Nigeria   |Mexico    |Indonesia |Korea     |China     |
> |----------|--------|--------|--------|--------|--------|--------|--------|--------|
> |Missing Ratio|9.56% | 0.0% |0.0% |0.45% |25.55% |0.32% |19.59% |10.75% |
>
> [1] Karpukhin, Vladimir, et al. "Dense Passage Retrieval for Open-Domain Question Answering." EMNLP (1). 2020.
>
> [2] Zhou, Yucheng, et al. "Towards robust ranker for text retrieval." Findings of the Association for Computational Linguistics: ACL 2023. 2023.
>
> [3] Wang, Junjun, et al. "Winning ClimateCheck: A multi-stage system with BM25, BGE-reranker ensembles, and LLM-based analysis for scientific abstract retrieval." Proceedings of the Fifth Workshop on Scholarly Document Processing (SDP 2025). 2025.
>
> [4] Santhanam, Keshav, et al. "Colbertv2: Effective and efficient retrieval via lightweight late interaction." Proceedings of the 2022 Conference of the North American Chapter of the Association for Computational Linguistics: Human Language Technologies. 2022.
>
> [5] Cao, Yong, et al. "Assessing cross-cultural alignment between ChatGPT and human societies: An empirical study." arXiv preprint arXiv:2303.17466 (2023).
>
> [6] Chiu, Yu Ying, et al. "CulturalBench: A robust, diverse and challenging benchmark for measuring LMs’ cultural knowledge through human-AI red-teaming." Proceedings of the 63rd Annual Meeting of the Association for Computational Linguistics (Volume 1: Long Papers). 2025.

---

> ### Author Response · Authors · 2025-11-20
> **Response to Reviewer nXd8 (3/3)**
>
> > Weakness 4: Potential data leakage due to image-only train/test split.
>
> Thank you for highlighting the critical concern regarding potential data leakage due to the image-only train/test split. We have double-checked the overlap across the splits at the document level and found the overlap percentages to be minimal: $\text{train} \cap \text{val}: 1.2\%$, $\text{train} \cap \text{test}: 5.1\%$, and $\text{val} \cap \text{test}: 8.3\%$. To fully address your concern and ensure the integrity of our evaluation, we performed strict de-duplication on the training and validation sets, guaranteeing that no identical positive documents  appear across the training, validation, testing splits. Following this de-duplication, we re-conducted the fine-tuning experiment. The results in the following table demonstrate the robustness of our model, showing that performance remains highly competitive. This confirms that our reported gains are due to the effectiveness of our proposed method, not data memorization.
>
> |dataset|MRR | P@1 | P@3| P@5| nDCG@1| nDCG@3| nDCG@5|
> |--|--|--|--|--|--|--|--|
> |CaCLIP|78.34 |65.42| 49.44| 39.50| 72.25| 75.22| 79.32|
> |CaCLIP (De-duplicated)|76.10|62.92|46.94|38.00|69.95|72.68|77.07|
>
> > Weakness 5: Missing statistics about positive vs. negative document ratios
>
> Thank you for highlighting the critical need for transparency regarding our evaluation setup. We agree that the ratio of positive to negative documents significantly impacts task difficulty and the meaningfulness of our results. To address this gap, we provide the precise positive/negative document ratios for each country set, confirming that the scenarios are far from trivial. Furthermore, we want to clarify that our existing negative set includes the documents that are already culturally similar but incorrect (e.g., documents from adjacent or geographically/linguistically related regions). To address your concern, we choose the last one as the hard negative one as the retrieval results and did experiments, the results are shown in the following table. **We have added this table to Appendix Table 6.**
>
> | Country   | Positive / Negative ratio  |
> |----------|----------------------------------|
> | India     | 57.07% / 42.93% |
> | Russia    | 54.42% / 45.58% |
> | Spain     | 48.17% / 51.83% |
> | Nigeria   | 34.47% / 65.53% |
> | Mexico    | 38.37% / 61.63% |
> | Indonesia | 41.26% / 58.74% |
> | Korea     | 69.59% / 30.41% |
> | China     | 56.12% / 43.88% |
>
>
>
> > Weakness 6 & Question 2: On Insufficient Analysis of Cultural Bias
>
> We agree that a deeper investigation into the causes of performance disparities is interesting. The primary goal of RAVENEA was to create the first reliable benchmark to enable and quantify the study of these biases in a RAG context. Our work provides the tool and the initial finding (e.g., underperformance on Nigerian contexts). A full causal analysis, such as disentangling annotation artifacts, knowledge source gaps (Wikipedia), and model biases, is a significant research endeavor in the future work, which we hope RAVENEA will inspire. However, our preliminary hypotheses are model training data. For instance, the volume and richness of English Wikipedia articles related to Nigerian cultural concepts are often less extensive than for Korean or Chinese concepts. Models pre-trained on web-scale data, which often over-represents Western and East Asian cultures, likely perpetuate these biases.
>
> > Question 1: The reason for the 8 countries
>
> Thank you for raising this question. To ensure the quality within limitation of the budget, the selection of eight countries from the initial thirty was guided by **the maximization of diversity and populations from various language branches** [7] (five were already present in CCUB). Our objective was to achieve broad geographic and cultural representation, sampling from four distinct continents: Asia (China, India, Indonesia, Korea), Africa (Nigeria), Europe (Russia, Spain), and Latin America (Mexico). **We clarify the reason in line 130.**
>
> [7] https://en.wikipedia.org/wiki/List_of_languages_by_total_number_of_speakers
>
> **Thank you again for your thoughtful and constructive feedback. We hope that our clarifications adequately address the questions or concerns you may have had. We would be grateful if you would kindly consider reassessing the score in light of our response.**

---

### Official Review · Reviewer_cUbB · 2025-11-01

**Soundness:** 3
**Presentation:** 3
**Contribution:** 4
**Rating:** 8
**Confidence:** 4

**Summary:**

The paper introduces RAVENEA, a retrieval-augmented, multimodal benchmark for cultural understanding across eight countries and eleven categories, built by pairing images from CVQA and CCUB with human-ranked Wikipedia passages and evaluating two tasks: culture-focused VQA (cVQA) and culture-informed captioning (cIC). The authors also propose RegionScore, a simple region-mention metric for cIC that correlates with human judgments better than standard captioning metrics, and Culture-Aware Contrastive (CAC) fine-tuning to adapt multimodal retrievers (producing CaCLIP/CaSigLIP) that notably improve retrieval and downstream performance, especially for smaller VLMs. This analysis highlights persistent cross-cultural gaps though there are diminishing returns for large models with RAG.

**Strengths:**

- Well-founded benchmark design with human relevance supervision. The pipeline uses GPT-4o captions → BM25 over ~6M English Wikipedia pages → human re-ranking with a clear, 3-question taxonomy (country association, topic alignment, explicit visual representation). The methodology of creation is also well outlined and easy to understand, with good documentation.

- Transparent quality control and strong agreement. There is multiple evidence that supervision is reliable and that the QA process was conducted at an acceptable standard, ensuring the reliability of the proposed benchmark and methodology. Caption checks show 92%/94% accuracies and agreement = 0.85. Moreover, the meta-check on annotations reports 98.2% acceptance and agreement = 0.83 and this prolongs the trend of faithful agreement between the data collection methodology line in line with how humans should have done.

- Substantive retriever improvements. The proposed CAC loss function works well in improving the performance of the new retriever when trained with the RAVENEA benchmark, signaling that the proposed auxiliary loss matters in fine-tuning the learning process for this specific use case. This method can potentially be extended into other tasks since the three losses are generic and widely used heuristics (contrastive, ranking, and intra-modal diversity regularization).

- Insightful findings on scale vs. retrieval and culture-specific gaps. RAG helps small VLMs substantially but shows diminishing or negative returns for the largest family members. The cross-country analysis shows persistent disparities, with weaker performance on specific regional subsets.

**Weaknesses:**

- Generalizability of the method. The paper emphasizes CAC and CaCLIP, which are trained and evaluated on RAVENEA. Without testing on external cultural datasets, it is difficult to claim general culture-aware retrieval superiority beyond “RAVENEA-specific specialization.” Evaluation on other datasets capturing similar concepts (e.g., CVQA, CCUB, CultureVLM, WorldCuisines, ALMBench) would strengthen the claim.

- The proposed RegionScore still measures lexical proxy rather than cultural semantic significance. By design, RegionScore is 1 if the caption contains the country or demonym, and 0 otherwise. This means it can reward trivial country mentions and miss high-quality culturally specific text without a country token. It limits what conclusions the cIC task can carry, and can even be harmful if a caption hallucinates an irrelevant country reference while describing a different cultural element.
--  The W/O RAG caption is: “Three individuals wearing vibrant traditional attire perform a dance outdoors.” This is generic. RegionScore = 0.
-- The With CaCLIP caption is: “A group of people wearing colorful traditional Igbo attire perform the Egedege dance at an event.” This is an excellent, culturally specific caption that captures deep nuance. However, it does not contain the word “Nigeria” or “Nigerian.” RegionScore = 0.
-- A hypothetical poor caption such as “This is a photo from Nigeria” would receive a higher RegionScore due to the occurrence of Nigeria. The metric is rewarding geotagging, not the cultural description it claims to measure. This is a conceptual flaw that undermines the cIC task, which is defined as assessing “captions that are sensitive to and incorporate cultural nuances.”

**Questions:**

Minor stuff:

- The paper mentions that the benchmark was derived from an initial pool of 30 countries but does not explain the rationale for narrowing down to eight. Clarifying this selection process would improve transparency and help readers assess dataset representativeness.
Line 126: “capabilities of of VLMs” → “of VLMs.”

- Several “under­standing / culturE uNdErstAnding” line breaks; clean hyphenation in camera-ready.

- Line 469: “incoming level (Nwatu et al., 2025)” likely should be “income level.”

- The introduction and Table 6 alternately say “over 10,000 documents” vs. “11,580 documents / 18,680 pairs.” Clarify whether “10k” is rounded or represents unique documents.

- Appendix A (LLM usage) and Section 2.2: consider reiterating that GPT-4o captions were not included in the released gold text; they were only used for retrieval augmentation. This is stated, but repeating it in the main text would prevent confusion.

---

> ### Author Response · Authors · 2025-11-20
> **Response to Reviewer cUbB (1/2)**
>
> Thank you for providing positive comments and the thoughtful feedback and attach our response as follows:
> > Weakness 1: Generalizability of the method.
> We appreciate this concern. Our response is twofold:
> 1. **Clarification of contribution:** The primary contribution of this paper is the RAVENEA benchmark itself, the first benchmark designed to evaluate multimodal retrieval-augmented cultural understanding. The CAC and CaCLIP methods were developed as strong baselines to demonstrate the utility of our benchmark and to provide a proof-of-concept for how culturally-aware retrieval can be effectively learned.
> We respectfully argue that reporting results on existing datasets such as cultureVLM, worldcuisine, detracts from the focus of this paper. **A direct evaluation of our RAG-focused method on those datasets is challenging because they are not structured for retrieval-augmented tasks.** Specifically, they lack the curated, human-ranked corpus of external documents paired with each image, which is the core component of RAVENEA. Our work was motivated by precisely this gap in existing resources.
> 2. **Results on CVQA and CCUB:** In Table 3, we present the results of two different downstream tasks. The image and questions utilized for cVQA and cIC tasks originate from the CVQA and CCUB datasets, respectively, and the documents are human-ranked documents. Notably, all the data used in downstream evaluation is excluded from the training process of CaCLIP. This experiment evaluate whether the culture-aware retrieval capabilities learned on RAVENEA training set can generalize to improve performance on a related downstream task.
>
> > Weakness 2: RegionScore still measures lexical proxy
>
> We are grateful to the your well-articulated analysis. In the context of LLM/VLM evaluation, a common and pragmatic starting point is to ground cultural concepts within a specific geo-political region. This approach is often seen in related works that aim to benchmark multicultural understanding [1, 2, 3], where regional origin is a primary axis for data stratification and analysis. We therefore designed RegionScore not as a definitive measure of "cultural fidelity," but as a scalable, automatic metric to quantify a model's ability to make a crucial first step: correctly identifying and naming the geo-cultural context.
>
> We agree that RegionScore is a lexical proxy. **However, we respectfully argue that it is not designed to measure caption quality in isolation.** Rather, it serves as a necessary **complementary supporter** to standard metrics like CLIPScore, which can evaluate the richness quality of the caption without ground truth captions. RegionScore and CLIPScore can be mutually essential. As you mentioned, the "Igbo" caption is superior. In our evaluation framework, this caption would receive a **high CLIPScore** due to its descriptive density and alignment with the visual content. Conversely, the trivial caption ("This is a photo from Nigeria") would receive a **low CLIPScore** because it lacks visual detail. As shown in Appendix Table 11, we noticed that the CLIPScore is closed to each other for most models, which means the richnesses of the captions are similar. And as shown in Table 1, standard metrics alone fail to capture cultural nuance. In conclusion, we do not claim RegionScore replaces semantic evaluation. Instead, we position it as a distinct, interpretable axis of evaluation that, when combined with standard metrics **as shown in Table 1 and Table 14**.
>
> > Question 1: The reason for the 8 countries
>
> Thank you for raising this concern. To ensure the quality within the budget constraints, the selection of eight countries from the initial thirty was guided by **the maximization of diversity and populations from various language branches**[4] (five of these countries were already present in CCUB). Our objective was to achieve broad geographic and cultural representation, sampling from four distinct continents: Asia (China, India, Indonesia, Korea), Africa (Nigeria), Europe (Russia, Spain), and Latin America (Mexico). **We clarify the reason in line 130.**
>
> [1] Liu, Fangyu, et al. "Visually grounded reasoning across languages and cultures." arXiv preprint arXiv:2109.13238 (2021).
>
> [2] Hershcovich, Daniel, et al. "Challenges and strategies in cross-cultural NLP." arXiv preprint arXiv:2203.10020 (2022).
>
> [3] Liu, Shudong, et al. "Culturevlm: Characterizing and improving cultural understanding of vision-language models for over 100 countries." arXiv preprint arXiv:2501.01282 (2025).
>
> [4] https://en.wikipedia.org/wiki/List_of_languages_by_total_number_of_speakers

---

> ### Author Response · Authors · 2025-11-20
> **Response to Reviewer cUbB (2/2)**
>
> > Minor stuff:
>
> We appreciate the valuable suggestions you provided. We would like to clarify that “over 10,000” refers to the rounded number, “11,580” represents the number of unique documents, and “18,680” denotes the unique image-document pairs. We have made the necessary modifications to the relevant section of the paper in accordance with your recommendations.
>
> **Thank you again for your thoughtful and constructive feedback. We hope that our clarifications adequately address the questions or concerns you may have had.**

---

### Author Response · Authors · 2025-11-30
**General response**

We thank all ACs and reviewers for the efforts and insightful comments. During the rebuttal, we actively addressed the reviewers' concerns through extra experiments and analysis.

**Before the leakage on Nov 27, 2025**:
- **Reviewer F5ZE** increased score $(6 \rightarrow 8)$ on **Nov 21st**.
- **Reviewer N9q4** maintained the positive score $(8 \rightarrow 8)$ on **Nov 26th**.
- The final average score is **6.5** (8/4/8/6).

We're glad that there is the consensus that our work offers a **timely contribution** to multimodal retrieval and culture (nXd8). Reviewers unanimously praised the **well-founded benchmark design** (cUbB), rigorous data curation (cUbB, F5ZE) (including **strong IAA** and **transparent quality control**), and **responsible ethical considerations** (F5ZE). The study features a comprehensive experimental design across **a wide range of models and tasks** (nXd8, F5ZE, N9q4, cUbB). Reviewers also highlight the **insightful findings** (cUbB, N9q4).

That said, reviewers also identified several important opportunities for improvement, addressed as follows (more details and updates in individual replies):
- **More statistics for the dataset**: we now provide two more tables (Table 6 and Table 8 in appendix) that show: (1) the missing documents ratio after the bm25 filtering, (2) the ratio of positive and negtive document per contry, (3) response distributions for the three cultural relevance annotation questions.
- **Retrieval qualty**: we now provide evidence that our annotation pipeline and criteria (more details in Appendix data statistics table 6) validate the high quality of our human-in-the-loop pipeline. What's more, during the rebuttal, we also provide the demo experiments to show the reason why it's very challenge for us to use other dense retrieval methods as the first stage.
- **Lexical-level evaluation metric (RegionScore)**: we provide the evidence that while CLIPScores indicate similar caption richness of RAG VLMs (Appendix Table 14), RegionScore correlates significantly better with human judgments in cultural awareness (Table 1). We therefore propose RegionScore as a distinct, interpretable axis of evaluation to be used alongside standard metrics, like CLIPScore.
- Expanded comparison to **new SOTA VLMs** (Qwen3VL family) now encompassing different retrievers in Table 3.
- Expanded comparison to **different retrieval settings (multi-paragraph, multi-documents)** with Qwen3VL and Gemma3 now in Table 13.
- Include the **reason for eight countries** (Section 2) by providing a link to the list of languages ranked by the number of speakers.

We thank all reviewers for their constructive feedback again; updates are highlighted in the manuscript with a blue color to make them easy to find.

---

### Meta-Review · Area_Chair_uzDg · 2026-01-05

**Summary:**

This paper makes a timely and substantive contribution by introducing RAVENEA, a carefully curated retrieval-augmented multimodal benchmark for visual cultural understanding, together with a thorough empirical study of retrieval-augmented generation (RAG) in vision–language models. Reviewer cUbB highlights the well-founded benchmark design with human-ranked Wikipedia evidence, transparent quality control, and high inter-annotator agreement, noting that the dataset construction pipeline is clearly documented and reliable. Reviewer nXd8 emphasizes the importance of addressing an underexplored intersection and commends the breadth of evaluation across seven retrievers and fourteen VLMs, spanning both open-source and proprietary models. Reviewer n9q4 underscores the clarity of the task formulation (cVQA and cIC), the scale and rigor of the annotations, and the diagnostic value of the benchmark for studying how models retrieve and use external cultural knowledge. Reviewer F5ZE further praises the comprehensive exploratory analysis across model scales and cultural dimensions, the release of culturally aware retrievers (CaCLIP/CaSigLIP) trained via Culture-Aware Contrastive Learning, and the attention to ethical dataset curation practices.

**Reviewer Concerns:**

- Reviewer cUbB and nXd8 both question the generalizability of the proposed CAC/CaCL retrievers beyond RAVENEA, recommending evaluation on additional cultural datasets to substantiate broader claims. Concerns about RegionScore recur across Reviewers cUbB and nXd8, who note that it relies on lexical country/demonym mentions and may reward superficial geotagging rather than deeper cultural semantics, potentially undermining conclusions drawn from the cIC task.
- Reviewer nXd8 also raises more fundamental methodological concerns, including the use of binary relevance annotations instead of graded judgments, lack of ablations or bias audits in the retrieval pipeline (e.g., BM25 vs. semantic retrievers), possible document-level leakage across splits, and insufficient reporting of positive/negative document ratios and hard negatives.
- Reviewer n9q4 points to limited domain coverage (eight countries, English Wikipedia) and a small-scale human evaluation for cIC, as well as a conservative RAG setup that may underestimate large-model performance.
- Reviewer F5ZE echoes concerns about dependence on GPT-4o captions and BM25 for candidate generation and the capped top-10 retrieval pool, which may limit recall.

**Reviewer Scores:**

- Reviewer cUbB: a positive rating (8) as discussion would reinforce that the benchmark quality and insights outweigh the acknowledged but non-fatal limitations (e.g., RegionScore and external generalization).

- Reviewer nXd8 (4): Likely would increase from 4 → 6, shifting to a weak accept after calibrating against other reviewers that most concerns are about rigor, reporting, and scope rather than fundamental flaws.

- Reviewer n9q4: stay positive, since their review is already well-aligned with consensus and frames limitations as future work rather than acceptance blockers.

- Reviewer F5ZE (increase from 4 to 6): as discussion would clarify that reliance on GPT-4o/BM25 is a design tradeoff and does not undermine the benchmark’s overall contribution.

---

### Decision · Program_Chairs · 2026-01-26

Accept (Poster)